**Data Availability Statement:** Experimental data are available in the Supplementary Information. Var

# Identification of novel PfEMP1 variants containing domain cassettes 11, 15 and 8 that mediate the *Plasmodium falciparum* virulence-associated rosetting phenotype

**Florence E. McLean[1]°, Brian R. Omondi[1]°, Nouhoum Diallo[1], Stanley Otoboh[1], Carolyne Kifude[1], Abdirahman I. Abdi[2], Rivka Lim[1], Thomas D. Otto[3], Ashfaq Ghumra[1], J. Alexandra Rowe[1] ***

**1** Institute of Immunology and Infection Research, School of Biological Sciences, University of Edinburgh, Edinburgh, United Kingdom, **2** KEMRI-Wellcome Trust Research Programme: Centre for Geographic Medicine Research Coast, Kilifi, Kenya, **3** Institute of Infection, Immunity and Inflammation, University of Glasgow, Glasgow, United Kingdom

° These authors contributed equally to this work.

\* alex.rowe@ed.ac.uk

## Abstract

*Plasmodium falciparum* erythrocyte membrane protein 1 (PfEMP1) is a diverse family of variant surface antigens, encoded by *var* genes, that mediates binding of infected erythrocytes to human cells and plays a key role in parasite immune evasion and malaria pathology. The increased availability of parasite genome sequence data has revolutionised the study of PfEMP1 diversity across multiple *P. falciparum* isolates. However, making functional sense of genomic data relies on the ability to infer binding phenotype from *var* gene sequence. For *P. falciparum* rosetting, the binding of infected erythrocytes to uninfected erythrocytes, the analysis of *var* gene/PfEMP1 sequences encoding the phenotype is limited, with only eight rosette-mediating PfEMP1 variants described to date. These known rosetting PfEMP1 variants fall into two types, characterised by N-terminal domains known as "domain cassette" 11 (DC11) and DC16. Here we test the hypothesis that DC11 and DC16 are the only PfEMP1 types in the *P. falciparum* genome that mediate rosetting, by examining a set of thirteen recent culture-adapted Kenyan parasite lines. We first analysed the *var* gene/PfEMP1 repertoires of the Kenyan lines and identified an average of three DC11 or DC16 PfEMP1 variants per genotype. *In vitro* rosette selection of the parasite lines yielded four with a high rosette frequency, and analysis of their *var* gene transcription, infected erythrocyte PfEMP1 surface expression, rosette disruption and erythrocyte binding function identified four novel rosette-mediating PfEMP1 variants. Two of these were of the predicted DC11 type (one showing the dual rosetting/IgM-Fc-binding phenotype), whereas two contained DC15 (DBLα1.2-CIDRα1.5b) a PfEMP1 type not previously associated with rosetting. We also showed that a Thai parasite line expressing a DC8-like PfEMP1 binds to erythrocytes to form rosettes. Hence, these data expand current knowledge of rosetting mechanisms and emphasize that the PfEMP1 types mediating rosetting are more diverse than previously recognised.

gene sequence data are previously published (reference 35, Otto et al., 2019) and are available at http://doi.10.5281/zenodo.3545835 and https://github.com/ThomasDOtto/varDB or from the Wellcome Sanger Institute FTP site (https://ftp.sanger.ac.uk/pub/project/pathogens/Plasmodium/falciparum/PF3K/varDB/). Sequence accession numbers for the Kenyan parasite lines are PFKE01/9106, ERS166380; PFKE02/9626, ERS166375; PFKE03/8383, ERS166384; PFKE04/10668, ERS166381; PFKE05/11014, ERS166382; PFKE06/10936, ERS166383; PFKE07/10975, ERS166385; PFKE08/9605, ERS166379; PFKE09/6816, ERS166374; PFKE10/11019, ERS166376; PFKE11/9775, ERS166378 and PFKE12/9215, ERS166377.

**Funding:** This work was funded by the Wellcome Trust (https://wellcome.org/) grant numbers 084226 (senior fellowship to JAR), 204052/Z/16/Z (PhD studentship to FEM), 218492/Z/19/Z (PhD studentship to RL and BRO) and 203077/Z/16/Z (AA funded by core Award to the KEMRI-Wellcome Trust Research Programme). The funders had no role in study design, data collection and analysis, decision to publish, or preparation of the manuscript.

**Competing interests:** The authors have declared that no competing interests exist.

## Author summary

Malaria is an infectious tropical disease caused by the parasite *Plasmodium falciparum* that kills more than half a million people every year, mostly young children in sub-Saharan Africa. Life-threatening episodes of malaria are characterised by huge numbers of parasitised red blood cells in the infected host, many of which bind to blood vessel walls and block blood flow, causing tissue damage and organ failure. Sometimes parasitised red cells also bind uninfected red cells to form clusters of cells called rosettes, which make the blockage of blood flow in vital organs even worse. Previous research has begun to decipher how parasitised red cells bind to uninfected red cells to form rosettes, but little is yet known about the process. Here we identify some new versions of the "sticky proteins" (adhesion molecules) that are made by malaria parasites and displayed on the surface of parasitised red cells to bring about rosette formation. The rosette-mediating adhesion molecules are members of a large family, and we identify here a few characteristic types within this family that mediate rosetting. This work is an important step towards the goal of understanding how malaria parasites form rosettes in order to develop preventions or treatments to reverse rosetting and reduce the number of people dying from severe malaria.

## Introduction

Rosetting is the binding of erythrocytes infected with *Plasmodium spp.* to uninfected erythrocytes during the asexual blood stage of malaria infection, and has been described for *P. falciparum*, *P. vivax*, *P. ovale* and *P. malariae*, as well as in murine and primate malaria parasite species (reviewed in [1]). Rosetting is a phenotypically variable property, with different parasite isolates having between zero and >90% of mature-infected erythrocytes in rosettes [2]. The rosetting phenotype is of particular interest in *P. falciparum* malaria, because studies in sub-Saharan Africa have consistently shown that parasite isolates from severe malaria cases have significantly higher rosetting levels than those from uncomplicated malaria cases [2–5], raising the possibility that rosetting contributes to disease pathogenesis. Subsequent work has shown that rosettes are stable under physiological blood flow shear forces and enhance microvascular obstruction caused by the sequestration of infected erythrocytes [6–9]. Furthermore, human erythrocyte polymorphisms that reduce the ability of *P. falciparum* to form rosettes, such as blood group O and the Complement Receptor One (CR1)-related Knops blood group, have been selected to a high frequency in malaria endemic regions and are associated with significant protection against severe disease [10–12], supporting the hypothesis that rosetting is a parasite virulence-associated phenotype.

Rosetting in *P. falciparum* infected erythrocytes begins at ~16–20 hours post invasion, a time-point corresponding to the appearance of parasite-derived variant antigens on the infected erythrocyte surface [13]. Specific members of the variant surface antigen family *P. falciparum* Erythrocyte Membrane Protein 1 (PfEMP1) are the best characterised rosette-mediating parasite adhesion molecules in *P. falciparum* [14–17], although there is some limited evidence to suggest that other VSA families such as the RIFINs and STEVORs may contribute to rosetting in some parasite genotypes [18,19]. PfEMP1 is encoded by *var* genes that can be classified into three main groups A, B and C and two intermediate groups B/A and B/C by their conserved upstream promoter sequences and genomic location [20]. These groups have functional significance as parasite isolates causing severe malaria predominantly transcribe group A and B/A *var* genes [21–25]. PfEMP1 variants have an N-terminal 'head structure'

composed of an N-terminal sequence (NTS) and Duffy Binding Like (DBL) domain followed by a Cysteine-rich InterDomain Region (CIDR)[26]. Downstream from the head structure, a variable number of further DBL and CIDR domains comprise the rest of the extracellular region. PfEMP1 DBL and CIDR domains have been classified using phylogenetic methods into seven (α, β, γ, δ, ε, ζ and pam) and five (α, β, γ, δ and pam) main classes respectively [26], which can be further divided into numerical subclasses [27]. Group A *var* genes encode DBLα1 domains, followed by a CIDRα1 or CIDRβ/γ/δ domain in their head structures, whereas group B and C *var* genes encode DBLα0 domains followed by CIDRα2–6 domains [26,27]. Domain subclasses frequently found together in tandem are called domain cassettes (DCs) [27].

PfEMP1 groups, domain cassettes and domain subclasses have been linked to specific host receptor binding phenotypes (reviewed in [28]), and the PfEMP1 domains that bind to host endothelial cell receptors such as CD36 [29], ICAM-1 [30] and endothelial cell protein receptor (EPCR) [31] are well-characterised. However, current understanding of the molecular interactions that cause rosetting is based on just eight PfEMP1 variants from six culture-adapted *P. falciparum* genotypes [14,15,17,32]. Seven of the eight characterised rosette-mediating PfEMP1 variants are encoded by group A *var* genes, and have a DBLα1 domain of subclass 1.5, 1.6 or 1.8 (Fig 1A). These domains are followed by a CIDRβ, γ or δ domain, completing DC11 in two of the characterised rosetting variants (IT4VAR60 and TM284VAR1), DC16 in three variants (3D7PF13_000, HB3VAR06 and MUZ12 VAR1) and DC16-like (similar to DC16, but with CIDRγ rather than CIDRδ) in two variants (IT4VAR09 and PAVARO) (Fig 1B). These domain subclasses and cassettes therefore define a rosetting-associated head structure (DBLα1.5/6/8-CIDRβ/γ/δ).

A single previously characterised candidate rosetting variant, TM180VAR1, is encoded by a group B/A *var* gene that lacks the rosetting-associated head structure [17]. This gene has a "DC8-like" head structure, with a DBLα2 domain followed by a CIDRα1.8 domain (Fig 1A), whereas DC8 usually has a CIDRα1.1 domain (Fig 1B) [27]. The unusual architecture of this variant, along with the observation that antibodies against the DBLα domain did not disrupt rosettes in this parasite line [17], means that the evidence supporting this as a rosette-mediating PfEMP1 type is weaker than for the other reported variants, and it remains possible that the TM180 parasite line forms rosettes via a PfEMP1-independent mechanism.

Due to the small number of characterised rosetting PfEMP1 variants to date, we currently lack a model to predict with confidence from parasite genomic sequence data which PfEMP1 variants mediate rosetting and which do not. In this study, we hypothesized that the rosetting phenotype in diverse *P. falciparum* isolates is encoded by either DC11 or DC16 PfEMP1 variants. We tested this hypothesis by firstly identifying PfEMP1 variants with the predicted rosetting head structure in the genomes of thirteen recently culture-adapted Kenyan *P. falciparum* lines, then selecting four of these lines for rosetting and characterising their *var* gene/PfEMP1 expression. We also examined whether the DC8-like candidate rosetting variant TM180VAR1 has erythrocyte-binding function. Our findings confirm the importance of DC11 and DC8-like variants in rosetting and identify another previously unrecognised PfEMP1 domain cassette (DC15) that binds to uninfected erythrocytes to mediate rosette formation.

## Results

### Selection of high rosetting (R+) Kenyan *P. falciparum* lines

Thirteen culture-adapted *P. falciparum* lines, originally isolated from children with *P. falciparum* malaria in Kenya, were studied [33,34]. The full *var* gene/PfEMP1 repertoire sequences of these parasites were published previously [35], and are shown in diagrammatic form in S1 Fig.

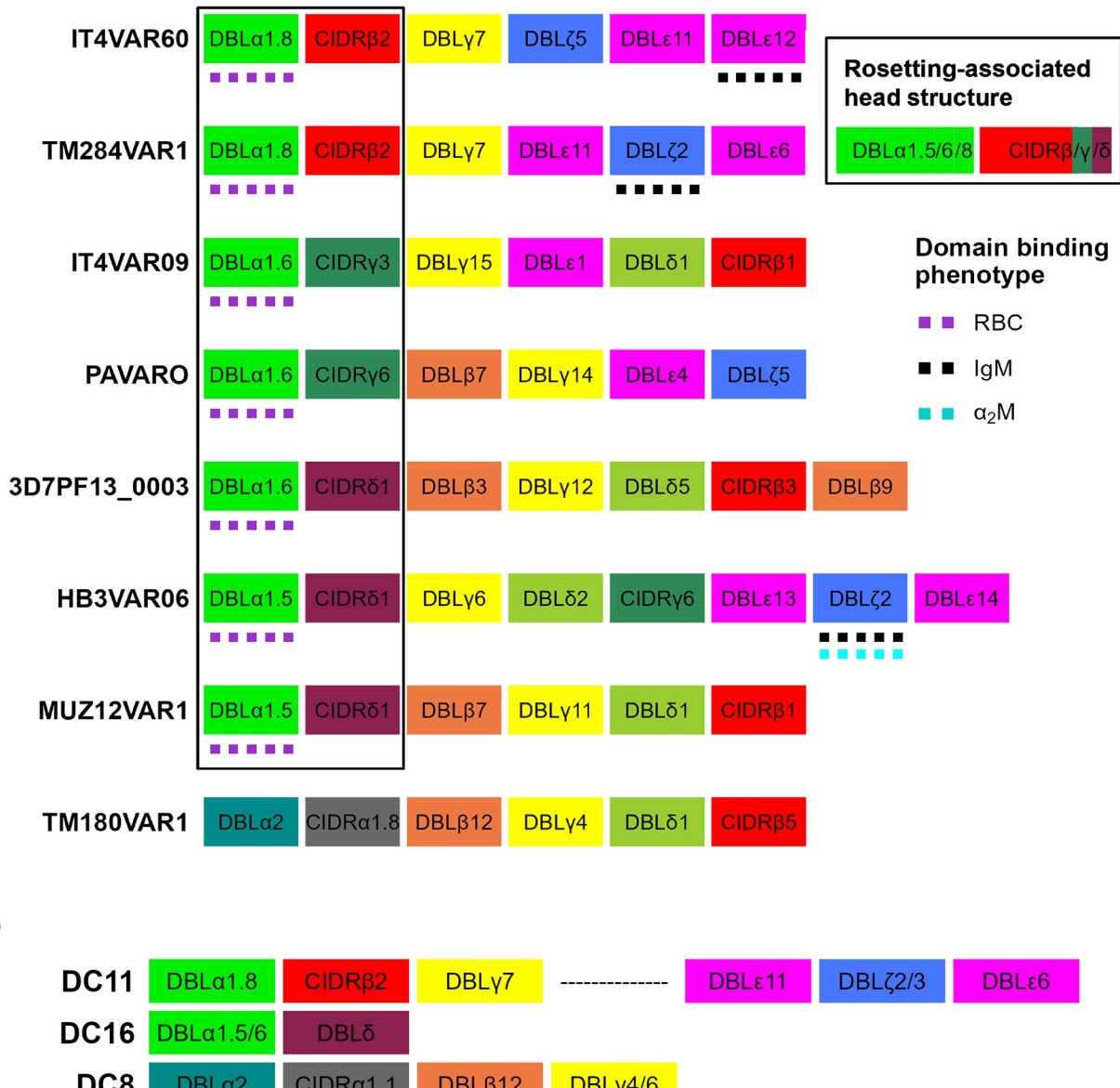

**Fig 1. Known rosette-mediating PfEMP1 variants and associated domain cassettes.** A) Diagram of the PfEMP1 domain architecture of known rosette-mediating variants [14–17]. The rosetting-associated head structure, DBLα1.5/6/8-CIDRβ/γ/δ, is boxed. Domains that bind erythrocytes (RBC) or serum proteins (IgM and alpha2Macroglobulin, α2M) are indicated. **B)** Diagram of domain cassettes (DCs) [as described by [27]] which are relevant to the rosetting PfEMP1 variants characterised previously.

On examining the repertoires, the number of PfEMP1 variants with the predicted rosetting-associated head structure (DBLα1.5/6/8-CIDRβ/γ/δ) ranged from 0 to 7 (median 3) per parasite line (Table 1). We cultured each of the Kenyan *P. falciparum* lines for at least two replication cycles *in vitro* and screened the mature-infected erythrocytes for spontaneous rosette formation by microscopy. Rosettes were detected in four of the parasite lines, at a high level in KE10 (>50% of infected erythrocytes in rosettes) and a low level in KE08, KE11 and PC0053 (2–10% of infected erythrocytes in rosettes). The absence of rosetting in the other parasite

**Table 1. The Kenyan *P. falciparum* parasite lines screened for rosetting *in vitro*.**

| Parasite line[#] | Pf3K designation[#] | Clinical syndrome[*] | Group A *var* genes[$] | Predicted rosetting head structure[†] | Rosettes seen *in vitro* | Human IgM-Fc binding[&] |
|---|---|---|---|---|---|---|
| 9106 | PFKE01 | IC | 7 | 1 | No | ND |
| 9626 | PFKE02 | UM | 15 | 3 | No | ND |
| 8383 | PFKE03 | UM | 9 | 2 | No | ND |
| 10668 | PFKE04 | IC, RD | 14 | 4 | No | ND |
| 11014 | PFKE05 | SMA | 6 | 2 | No | ND |
| 10936 | PFKE06 | UM | 11 | 6 | No | ND |
| 10975 | PFKE07 | SMA | 23 | 7 | No | ND |
| **9605** | **PFKE08** | **IC** | **8** | **4** | **Yes** | **No** |
| 6816 | PFKE09 | IC | 6 | 0 | No | ND |
| **11019** | **PFKE10** | **IC** | **10** | **3** | **Yes** | **Yes** |
| **9775** | **PFKE11** | **UM** | **13** | **3** | **Yes** | **No** |
| 9215 | PFKE12 | RD | 12 | 3 | No | ND |
| **9197** | **PC0053** | **IC** | **8** | **3** | **Yes** | **No** |

[#]The name of each parasite line according to the KEMRI-Wellcome Trust Laboratory parasite naming system [33,34] and its designation in the Pf3k database [35].

[*]The clinical syndrome of the patient from whom the parasite isolate was collected. IC, impaired consciousness; UM, uncomplicated malaria: RD, respiratory distress; SMA, severe malarial anaemia.

[$]The total number of group A *var* genes in each parasite's genome.

[†]The number of *var* genes in each parasite's genome that encode the rosetting-associated head structure DBLα1-CIDRβ/γ/δ.

[&]Determined by immunofluorescence assay/flow cytometry of live infected erythrocytes with an anti-human IgM (μ chain) antibody. ND, not done.

lines, despite their genomes containing group A *var* genes encoding PfEMP1 variants with the rosetting head structure, is not surprising because previous work has shown that parasites tend to switch off group A *var* genes when adapted to culture [36]. The four rosette-positive lines were grown in continuous culture and selected using density gradient methods to generate isogenic pairs of high rosetting (R+, >60% of infected erythrocytes in rosettes) and low rosetting (R-, <2% of infected erythrocytes in rosettes) parasite lines for each genotype. Parasite line KE11R+ was resistant to selection for high rosetting, possibly due to frequent *var* gene switching. Therefore, further study of KE11R+ was carried at a rosette frequency of ~15–30% of infected erythrocytes in rosettes.

## IgM-binding, trypsin-sensitivity and heparin-sensitivity of rosetting in the Kenyan R+ parasite lines

Previous work has shown that some *P. falciparum* rosetting lines bind the Fc region of human IgM, and this promotes rosette formation, possibly by clustering PfEMP1 molecules to increase binding avidity [37–39]. The dual rosetting and IgM-Fc binding phenotype is strongly associated with severe malaria [40,41] and the known PfEMP1 variants that mediate this dual binding are mostly of the DC11 type (Fig 1) [17,37,38,42]. Due to the presence of human serum in the parasite culture medium, IgM-binding can be detected by immunostaining of live infected erythrocytes taken directly from culture. Hence, the four R+ lines were assessed for IgM-binding by staining with anti-human IgM antibodies. Infected erythrocytes of the KE10R+ parasite line were IgM positive, whereas the other three R+ lines were IgM negative (Fig 2A and Table 1).

PfEMP1 is characteristically highly sensitive to trypsin digestion [17,43], whereas RIFINs and STEVORS, which have also been implicated in rosetting, are not [44,45]. Therefore, to determine whether PfEMP1 was the likely rosette-mediating parasite adhesin in the Kenyan

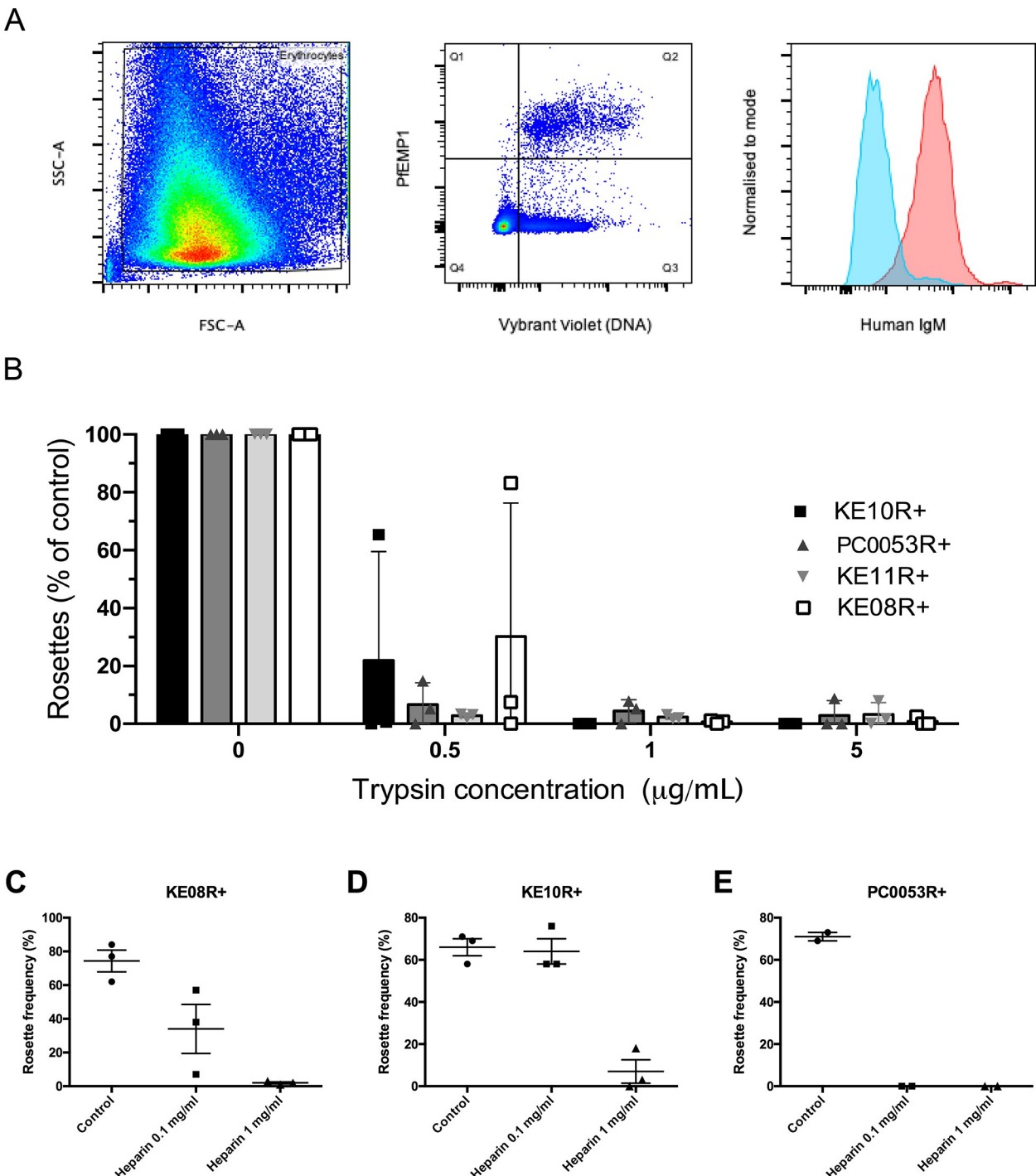

**Fig 2. Properties of the new Kenyan rosetting lines.** A) IgM binding by KE10R+ infected erythrocytes detected by flow cytometry. Forward and side scatter were used to gate on erythrocytes and exclude debris (left panel) and PfEMP1-expressing infected erythrocytes were detected by staining with 20µg/ml of polyclonal rabbit IgG against NTS-DBLα of KE10VAR_R1 followed by 1/1000 dilution of Alexa Fluor 647-conjugated goat anti-rabbit IgG secondary antibody and 1/2500 dilution of Vybrant DyeCycle Violet (middle panel Q2). IgM staining of the Q2 cell population was detected with 1/1000 dilution of an Alexa Fluor 488-conjugated goat anti-human IgM heavy chain antibody (red). The negative control (blue) was parasites grown in IgM-depleted medium and stained with the same antibodies. Results are representative of two independent experiments. B) Effect of low-dose trypsinisation on rosetting. Purified infected erythrocytes were treated with 0.5, 1 or 5 µg/ml of trypsin for 5 mins and the rosette frequency relative to a control with no added enzyme was calculated. The mean and standard deviation from three independent experiments per parasite line is shown. The rosette frequency of the untreated control was between 46%-81% (KE10R+), 45%-89% (PC0053R+), 16%-30% (KE11R+) and 61%-91% (KE08R+). C-E) Effect of heparin on rosetting. The mean and standard error from two or three independent experiments per parasite line is shown.

rosetting lines, infected erythrocytes were treated with low dose trypsin. Rosetting in all four lines was abolished by treatment with 1μg/ml trypsin for five minutes, consistent with a PfEMP1-dependent rosetting mechanism (Fig 2B). Many known PfEMP1-dependent rosetting parasites are sensitive to rosette disruption by heparin [46], and this property was also shown by the Kenyan lines (Fig 2C–2E).

## *var* gene profiling to identify predominant rosetting-associated PfEMP1 variants

To identify the predominant *var* genes specifically transcribed in the R+ parasite lines, we carried out DBLα tag profiling of the R+/R- isogenic pairs [14,47]. The full *var* gene sequence for each predominant tag was retrieved from the Pf3k database [35] or determined by extension PCR and sequencing. For clarity, the predominant rosette-specific *var* gene identified in each parasite line is referred to here as '*parasite line name var_r1*' with the full name of each gene from the Pf3K database also given on first use. *var* gene profiling of KE10R+ and KE10R- parasite lines showed that *pfke10var_r1* (*PX0203.g54*) encoding a DC11 PfEMP1 variant was the predominant transcript expressed only in the rosetting line (Figs 3 and 4). Parasite lines PC0053R+ and KE11R+ both expressed two predominant transcripts. However, in each case, only one was exclusively expressed by the R+ line (Fig 3). Therefore, *pc0053var_r1 (PC0053-C. g687)* and *pfke11var_r1 (PFKE11.g448)*, encoding PfEMP1 variants containing DC15 (DBLα1.2-CIDRα1.5b), were identified as the candidate rosette-mediating adhesins (Fig 4). This head structure has not previously been associated with rosetting. *var* gene profiling of the KE08R+ and KE08R- parasite lines identified two predominant transcripts in the R+ population, *pfke08var_r1 (PFKE08.g502)* and *pfke08var_r2 (PFKE08.g501)* (Fig 3). Expression of *pfke08var_r2* was not detected in the R- population, whereas *pfke08var_r1* was detected at a low level (5% of transcripts). *pfke08var_r1* encodes a DC11 rosetting-associated head structure, whereas *pfke08var_r2* is a group B/A *var* gene, encoding DC8 (DBLα2-CIDRα1.1) (Fig 4). DC8 PfEMP1 variants are usually associated with binding to human brain endothelial cells via EPCR [48], and not with rosetting, with the exception of the unusual putative rosetting variant TM180VAR1, which has a "DC8-like" architecture [17]. Therefore, in this case, both *pfke08-var_r1* and *pfke08var_r2* were considered to be potential rosette-mediating PfEMP1 variants and were investigated further.

## Staining of infected erythrocytes for surface expression of candidate rosetting-associated variants

We next raised antibodies to the candidate rosetting PfEMP1 variants identified above, to determine whether the specific PfEMP1 variants were present on the surface of live infected erythrocytes. The NTS-DBLα region of each variant was expressed as a recombinant protein in *E. coli* (S2 Fig) and used to immunise rabbits to generate polyclonal antibodies against PfEMP1 as described previously [17,49]. The antibodies were used in indirect immunofluorescence and flow cytometry and non-immunised rabbit IgG or IgG raised to an irrelevant PfEMP1 (NTS-DBLα of HB3VAR03, a non-rosetting variant that binds to human brain endothelial cells [50]) were used as negative controls. The gating strategy to identify the mature pigmented-trophozoite and schizont stage infected erythrocytes as a DNA/RNA high population is shown in S3 Fig.

In each case, the IgG against the candidate rosetting PfEMP1 variant stained live rosetting infected erythrocytes of the respective parasite line at a frequency comparable to the rosette frequency, whereas the isogenic R- lines were negative (Fig 5). Low dose trypsinisation removed PfEMP1 (S4 Fig), consistent with the absence of rosetting after trypsin treatment

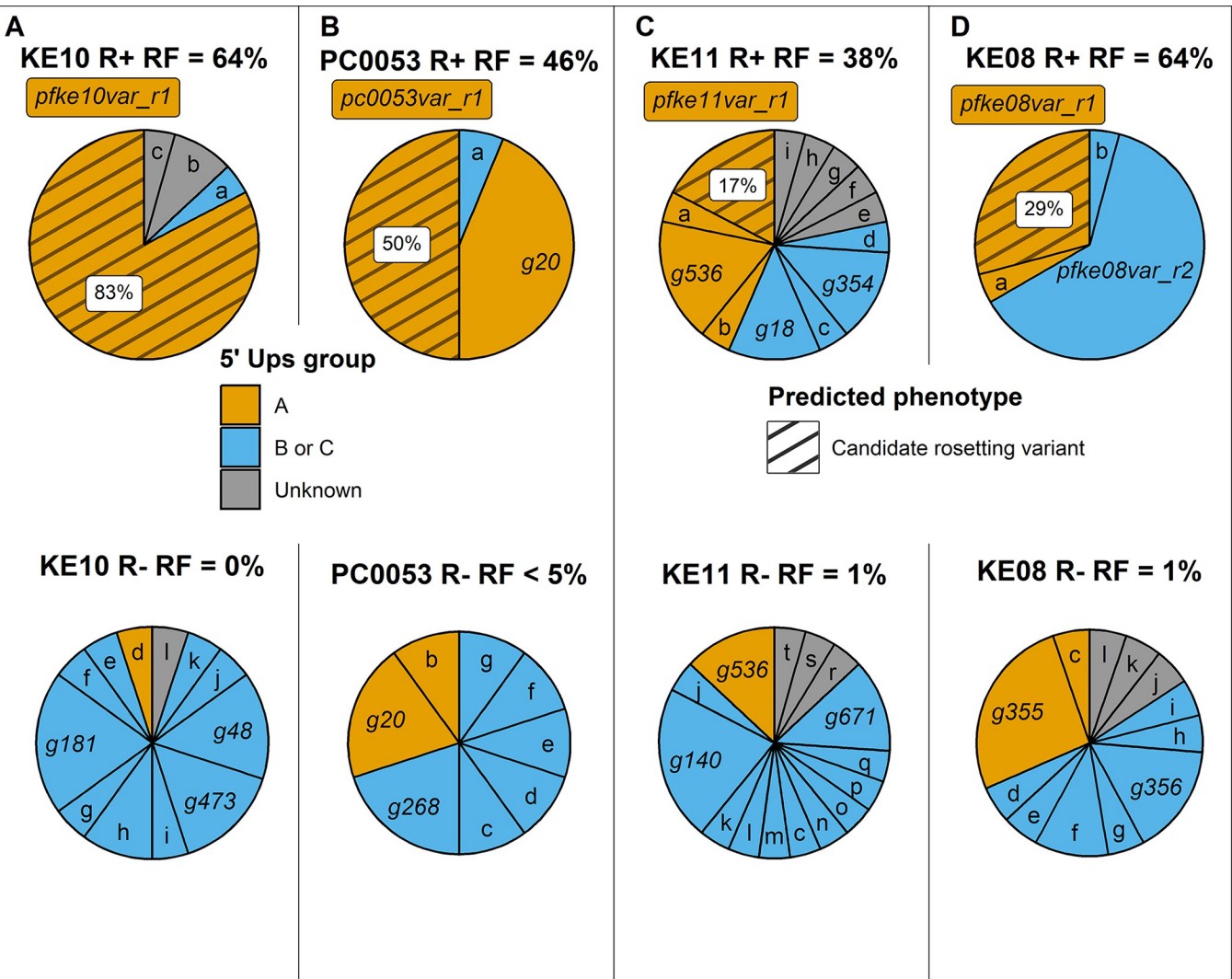

**Fig 3. DBLα expression tag profiling of the new Kenyan rosetting lines.** DBLα tag expression profiles showing the proportion each gene tag contributed to the overall expressed tag profile for each parasite population. *var* gene tags encoding candidate rosetting variants (hatched; percentage of all tags given in white box) are identified as predominant genes expressed in R+ parasites but not/rarely in isogenic R- parasites. Other large pie slices are identified by gene (g) number for each parasite line, and small slices are given letters, with the full information for each gene given in S1 Table. *var* gene groups are identified by colour. Slices labelled as "unknown" are from sequence tags that did not map back to an assembled *var* gene in that parasite's repertoire [35]. The rosette frequency (RF) in the cycle in which the RNA was collected is shown for each parasite line. Results are representative of two independent experiments.

(Fig 2), supporting the identification of PfEMP1 as the likely rosette-mediating adhesin. Unexpectedly, in the KE08R+ line ~90% of mature infected erythrocytes stained positively with both the KE08VAR_R1-specific rabbit IgG and the KE08VAR_R2-specific IgG (Fig 5E and 5G). This suggests either antibody cross-reactivity or the possibility that both PfEMP1 variants are expressed on the same infected erythrocytes. Usually only one PfEMP1 type is expressed per infected erythrocyte, due to mutually exclusive expression of *var* genes [51]. However, one previous example of a parasite line simultaneously expressing two different PfEMP1 variants on the same infected erythrocyte has been described [52]. To examine KE08R+ in more detail, we raised antibodies against the N-terminal CIDR domains of the two predominantly expressed PfEMP1 types and repeated the immunofluorescence staining. Once again, the polyclonal rabbit IgG raised against KE08VAR_R1 CIDR and KE08VAR_R2 CIDR both stained a

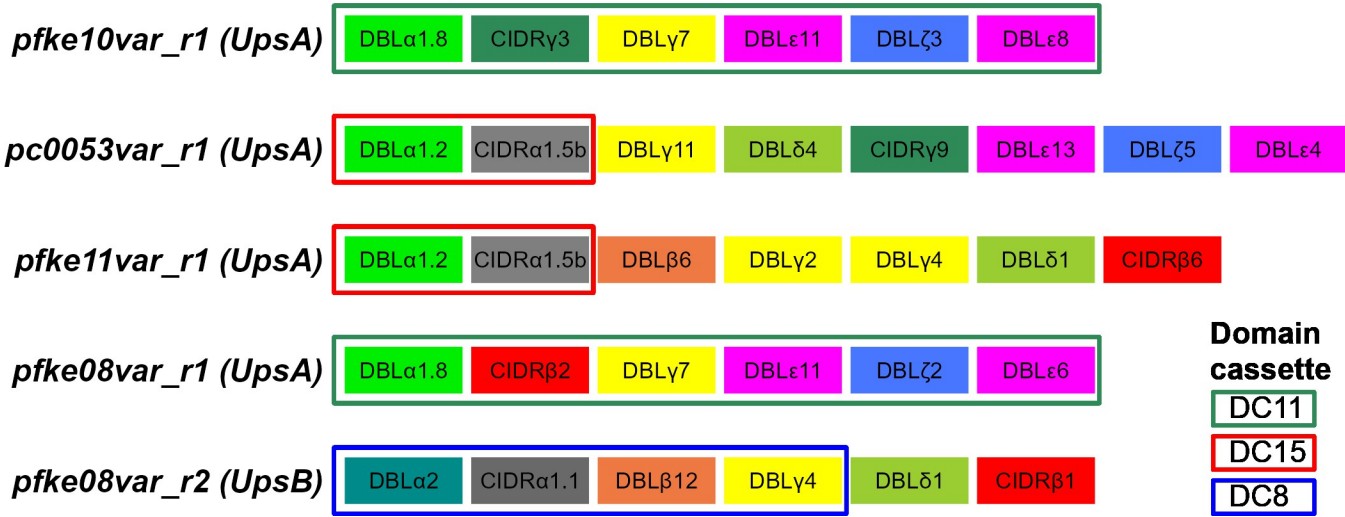

**Fig 4. Domain architecture of the candidate rosette-mediating PfEMP1 variants.** Domain cassettes are as described by Rask et al [27]. The alternative names for these genes in the Pf3k database [35] are as follows: *pfke10var_r1/PX0203.g54; pc0053var_r1/PC0053-C.g687; pfke11var_r1/ PFKE11.g448; pfke08var_r1/ PFKE08.g502; pfke08var_r2/ PFKE08.g501.*

similar major population of infected erythrocytes in the rosetting line, consistent with expression of both variants on the surface of infected erythrocytes (Fig 5I and 5K).

## Rosette disrupting activity of anti-PfEMP1 IgG

Next, we assessed the ability of the rabbit IgG raised against the candidate rosetting variants to disrupt rosettes in their respective lines. IgG from antisera raised to the NTS-DBLα domain of an irrelevant PfEMP1 variant (HB3VAR03) and IgG purified from the sera of a non-immunised rabbit were used as negative controls. The KE11R+ line was not included in these assays, due to the difficulty in maintaining a sufficiently high level of rosetting. For parasite lines KE10R+ and PC0053R+, rosetting was completely abolished by 10-100μg/ml of rabbit IgG to the relevant PfEMP1 NTS-DBLα domain (Fig 6A and 6B), supporting the identification of KE10VAR_R1 (DC11) and PC0053VAR_R1 (DC15) as rosette-mediating PfEMP1 types. In the KE08R+ line, rosettes were abolished by 10μg/ml of IgG to the NTS-DBLα domain of KE08VAR_R1 (DC11), but there was no effect on rosetting of IgG to KE08VAR_R2 (DC8) (Fig 6C), despite these antibodies recognising the surface of KE08R+ infected erythrocytes (Fig 5G). This suggests that it is the DC11 variant KE08VAR_R1 and not the DC8 variant KE08VAR_R2 that is the rosette-mediating adhesion molecule in the KE08R+ parasite line.

## Functional assays to detect PfEMP1 binding to uninfected erythrocytes and IgM

Previous studies have identified the NTS-DBLα domain as the erythrocyte-binding region of rosette-mediating PfEMP1 variants [14,15,17,32]. To examine the erythrocyte binding function of the candidate rosetting variants, NTS-DBLα and CIDR recombinant proteins were incubated with uninfected erythrocytes and bound protein detected by flow cytometry. The NTS-DBLα domain of the well-characterised rosetting variant IT4var60 [15,17] was included as a positive control, alongside negative controls which were NTS-DBLα domains from human brain endothelial cell binding PfEMP1 variants that are expressed by parasite lines that are known to be non-rosetting [50].

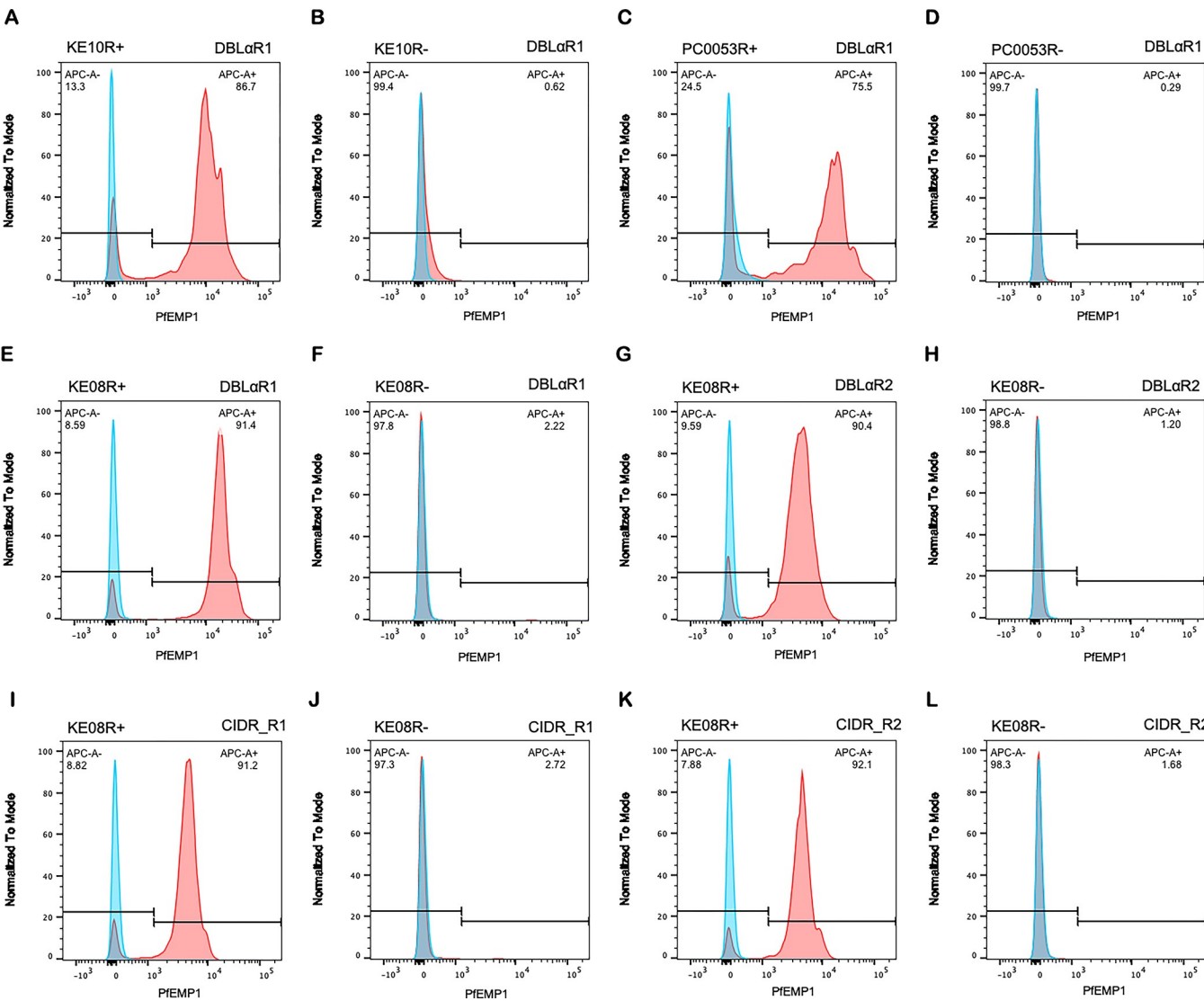

**Fig 5. Staining of the surface of live infected erythrocytes with PfEMP1 antibodies.** Fluorescence intensity histograms of rosetting (R+) and non-rosetting (R-) parasite lines stained with rabbit IgG against rosetting PfEMP1 variants (red) or negative control (rabbit IgG against the NTS-DBLα domain of an irrelevant PfEMP1 variant, HB3VAR03, blue). A) parasite line KE10R+ with antibodies to KE10VAR_R1 NTS-DBLα. B) parasite line KE10R- with antibodies to KE10VAR_R1 NTS-DBLα. C) parasite line PC0053R+ with antibodies to PC0053VAR_R1 NTS-DBLα. D) parasite line PC0053R- with antibodies to PC0053VAR_R1 NTS-DBLα. E) parasite line KE08R+ with antibodies to KE08VAR_R1 NTS-DBLα. F) parasite line KE08R- with antibodies to KE08VAR_R1 NTS-DBLα. G) parasite line KE08R+ with antibodies to KE08VAR_R2 NTS-DBLα. H) parasite line KE08R- with antibodies to KE08VAR_R2 NTS-DBLα. I) parasite line KE08R+ with antibodies to KE08VAR_R1 CIDR. J) parasite line KE08R- with antibodies to KE08VAR_R1 CIDR. K) parasite line KE08R+ with antibodies to KE08VAR_R2 CIDR. L) parasite line KE08R- with antibodies to KE08VAR_R2 CIDR. PfEMP1 was detected with 20μg/ml rabbit IgG against PfEMP1 and 1/1000 dilution of Alexa Fluor 647-conjugated goat anti-rabbit IgG secondary antibody. Gates shows the percentage of mature infected erythrocytes positive for the variant. The rosette frequency at the time of staining was 70–90% for the R+ lines and <2% for the R- lines, and the percentage of positive staining infected erythrocytes (APC-A+, top right corner of each histogram) closely matched the rosette frequency in each parasite line. Results are representative of at least two experiments for each parasite line.

The NTS-DBLα domains of the candidate rosetting variants KE10VAR_R1 (DC11), PC0053VAR_R1 (DC15), KE11VAR_R1 (DC15) and KE08VAR_R1 (DC11) all bound to uninfected erythrocytes, confirming their identification as rosette-mediating variants (Fig 7A–7F). Neither the NTS-DBLα or CIDR domain from KE08VAR_R2 bound to erythrocytes, consistent with the above suggestion that KE08VAR_R2 is not a rosette-mediating variant (Fig 7A

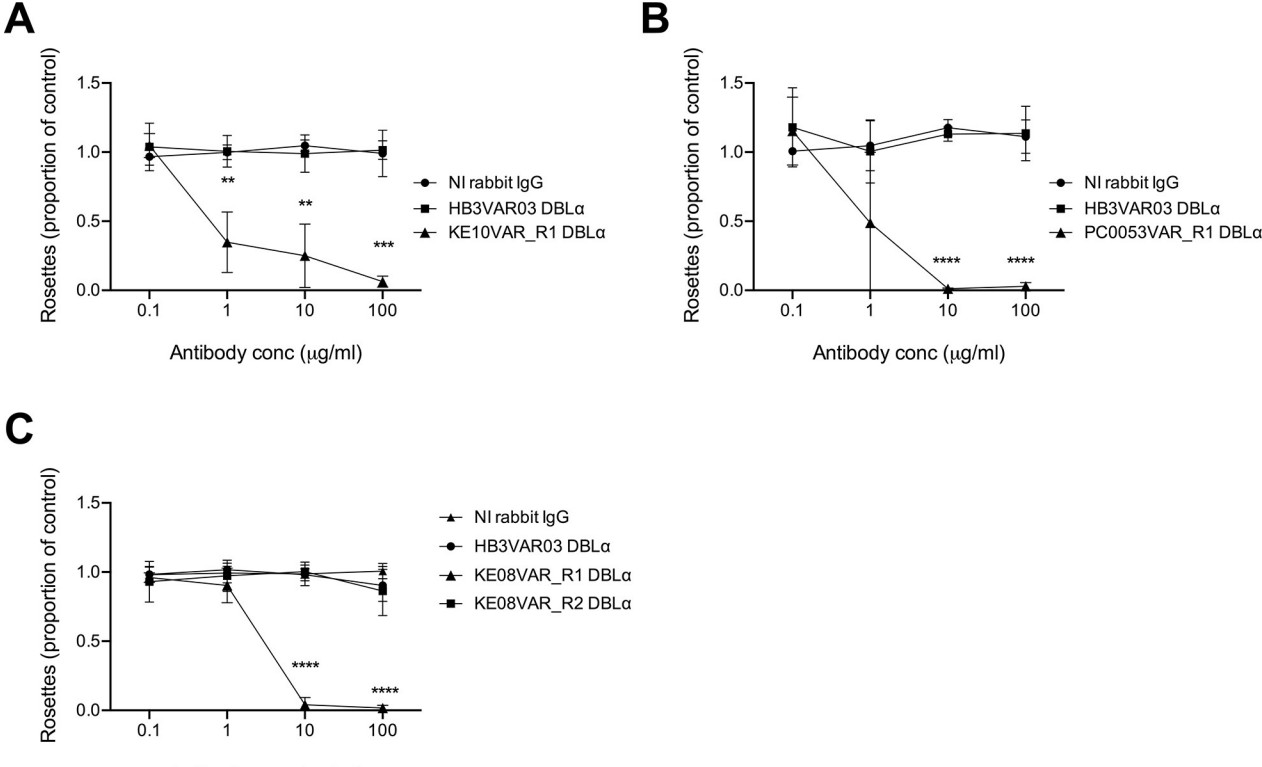

**Fig 6. Rosette disruption with rabbit IgG raised against candidate rosetting PfEMP1 variants.** Antibodies to the candidate rosette-mediating PfEMP1 variant NTS-DBLα domains were tested in rosette-disruption assays over a range of concentrations from 0.1–100 µg/mL. The rosette frequency in the presence of antibody is shown as the proportion of the control value with no added antibody. IgG from a non-immunised (NI) rabbit and from a rabbit immunised with the NTS-DBLα domain of a non-rosetting PfEMP1 variant HB3VAR03 were used as negative controls. Data represent three independent experiments with the mean and standard deviation of the three experiments shown. A) parasite line KE10R+; the rosette frequency of the no antibody control ranged from 71%-87%. B) Parasite line PC0053R+; the rosette frequency of the no antibody control ranged from 49%-60%. C) Parasite line KE08R+; the rosette frequency of the no antibody control ranged from 69%-92%. Data were analyzed by two-tailed paired t tests corrected for multiple comparisons with the Holm-Sidak method ** P<0.01, *** P<0.001, **** P<0.0001.

and 7G). The CIDR domains of KE08VAR_R1 and KE11VAR_R1 also did not bind erythrocytes, consistent with previous data showing NTS-DBLα to be the major erythrocyte-binding domain of rosette-mediating PfEMP1 variants (Fig 7A) [14,32,53].

For the IgM Fc-binding rosetting line KE10R+, the domains downstream of the head structure were expressed as recombinant proteins and tested for human IgM binding by ELISA. Previous work has identified either the ultimate DBLε or penultimate DBLζ domain as the IgM-binding region of PfEMP1 (Fig 1) [37–39,54]. Consistent with this, DBLζ3 was identified as the IgM-binding domain of KE10VAR_R1 (Fig 8).

## The "DC8-like" variant TM180VAR1 mediates rosetting

The rosetting line TM180R+ expresses the group B/A *var* gene *tm180var1*, which lacks the rosetting associated head structure (Fig 1) [17]. To test the hypothesis that TM180R+ forms rosettes via a variant surface antigen other than PfEMP1, TM180R+ was treated with low-dose trypsin as described above. TM180R+ rosetting was abolished by 5 µg/ml of trypsin, suggestive of a PfEMP1-mediated rosetting mechanism rather than an alternative RIFIN- or STEVOR-mediated mechanism (Fig 9A). The TM180VAR1 NTS-DBLα recombinant protein was tested

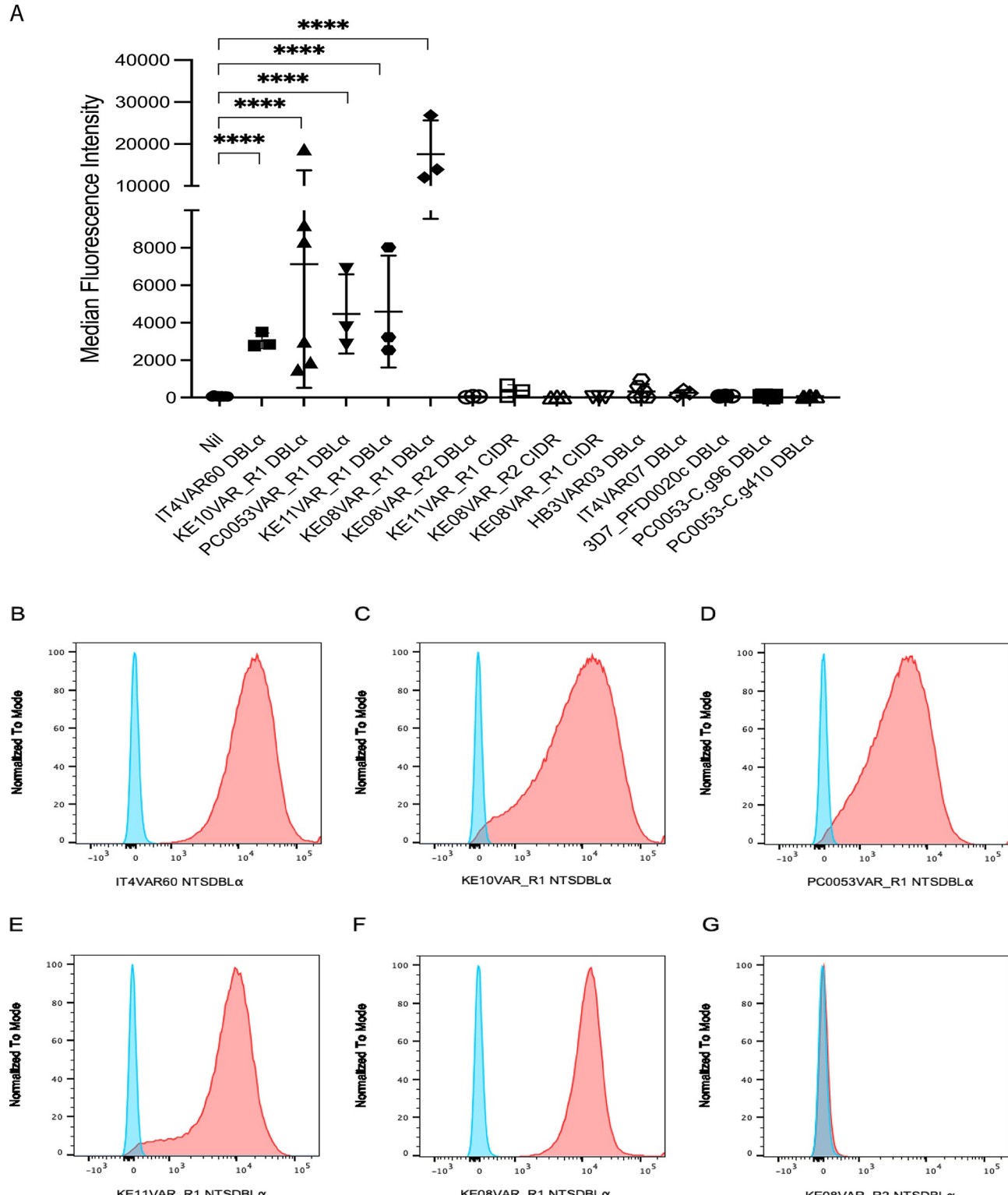

**Fig 7. Binding of PfEMP1 recombinant proteins to erythrocytes.** A) Recombinant proteins were incubated with uninfected erythrocytes and bound protein detected with specific antibodies to each domain. The positive control was the NTS-DBLα domain of the well-characterised rosetting variant IT4VAR60 [15,17] and negative controls were the NTS-DBLα domains from human brain endothelial cell binding PfEMP1 variants HB3VAR03 (PFHB3_130080100), IT4VAR07 (PFIT_060036700), 3D7_PFD0020c (PF3D7_0400400), PC0053-C.g96 and PC0053-C.g410 that are known to be non-rosetting [50]. The mean and standard deviation of the Alexa Fluor 488 median fluorescence intensity from at least three independent experiments per

protein is indicated. Data were log transformed and analysed by one way ANOVA with Dunnett's multiple comparisons test compared to the "Nil" (no added protein) control. **** P<0.0001. B-G) Example Alexa Fluor 488 fluorescence intensity histograms of recombinant PfEMP1 domains bound to erythrocytes and detected by indirect immunofluorescence (red) compared to a non-rosetting NTS-DBLα domain negative control (blue). The domain tested is indicated below each histogram.

in the erythrocyte binding flow cytometry assay, and showed unequivocal binding to uninfected erythrocytes, confirming its role in rosetting (Fig 9B).

## KE08R+ binds to ICAM-1 and EPCR as well as forming rosettes

The simultaneous expression and surface display of KE08VAR_R1 and KE08VAR_R2 shown above raise the possibility that the KE08R+ parasite line might have an unusual adhesion phenotype. KE08VAR_R2 contains a CIDRα1.1 domain that would be predicted to bind EPCR [30,48]. Static binding experiments showed that KE08R+ infected erythrocytes bound EPCR and ICAM-1 (Fig 10). The latter was unexpected because the DBLβ12 domain in KE08VAR_R2 is predicted to bind gC1QR rather than ICAM-1 [30,48,55]. Notably, significantly higher binding to EPCR and ICAM-1 was seen when rosettes were disrupted with rabbit IgG to KE08VAR_R1 prior to the assay, showing that rosettes impede binding to other receptors in adhesion assays, as shown previously [56]. These data indicate that, by expressing two PfEMP1 variants simultaneously, KE08R+ infected erythrocytes exhibit an unusual "triple" virulence-associated adhesion phenotype—rosetting, ICAM-1 binding and EPCR-binding.

## Presence of rosetting-associated motifs in the novel rosetting PfEMP1 variants

Having identified four new rosette-mediating PfEMP1 variants, we examined a multiple sequence alignment of known rosetting and non-rosetting NTS-DBLα amino acid sequences to determine if previous motifs [4] and homology blocks (HB) [57] associated with high rosetting were present. The motif "MFKPNKDKVEG" that contains HB219 [57] and the adjacent high rosetting "H3" motif [4] was seen in all of the newly described variants and this sequence is over-represented in rosetting compared to non-rosetting variants (green box, Fig 11). The other Normak et al [4] high rosetting motifs (H1 and H2) were not found. Two other motifs in

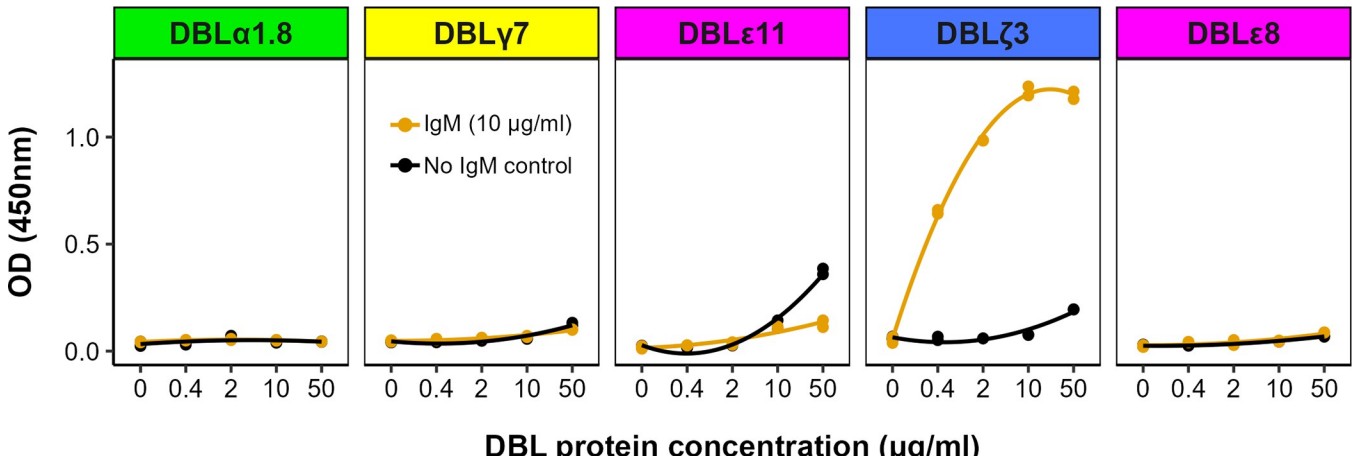

**Fig 8. The DBLζ3 domain of PFKE10VAR_R1 binds human IgM.** IgM was coated onto the wells of an ELISA plate and individual PfEMP1 recombinant protein domains were added at four different 5-fold dilutions. Wells with no IgM were included as negative controls. Binding was detected using an HRP-conjugated anti-His tag antibody with TMB substrate and the absorbance read at 450nm. The domains are shown from N- to C-terminal and the panels are colour-coded as previously described [27]. One representative experiment is shown out of at least two performed for each domain.

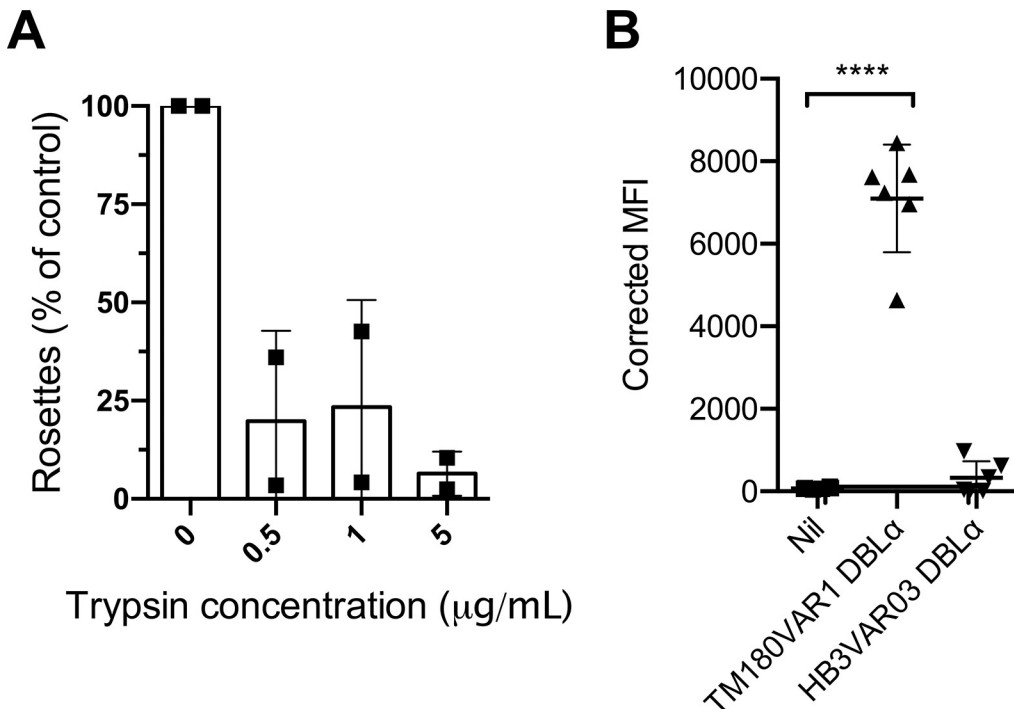

**Fig 9. The NTS-DBLα domain of TM180VAR1 mediates rosetting.** A) Effect of low dose trypsinisation on TM180R+. Infected erythrocytes were treated with 0.5, 1 or 5 µg/ml of trypsin for 5 mins in each experiment and the rosette frequency relative to a control sample with no added enzyme was calculated. Data represent two independent experiments with the mean indicated by bar height and the standard deviation shown. The rosette frequency in the untreated controls were 72% and 61% respectively. B) Corrected Median Fluorescence Intensities (MFI) of uninfected erythrocytes in binding experiments with recombinant TM180VAR1 NTS-DBLα and the negative control HB3VAR03 NTS-DBLα compared to "Nil" no added protein control. Data are from six independent experiments and the mean and standard deviation are shown. Significant differences by one-way ANOVA with Dunnett's multiple comparisons test compared to the "Nil" negative control are indicated. ****P<0.0001.

the central region of NTS-DBLα "YKDGSG" (red box, Fig 11) and "TCKAPQDAN" (blue box, Fig 11) were found in the newly identified variants and are strongly over-represented in the rosetting variants overall.

### Rosetting NTS-DBLα domain frequency in global *P. falciparum* isolates

The frequency of rosetting-associated NTS-DBLα domains (1.2, 1.5, 1.6, 1.8) within and outside of their respective domain cassettes in the Pf3k database [35] was investigated. NTS-DBLα1.8 occurred more than 600 times in the genomes of 714 *P. falciparum* isolates, usually linked to a CIDRβ/γ/δ giving DC11 (Fig 12A). Similar results were seen for NTS-DBLα1.5 and NTS-DBLα1.6, which almost always occur within DC16, suggesting limited recombination with *var* genes encoding other domain cassette types in these cases. NTS-DBLα1.2 showed a different pattern, with less than half of NTS-DBLα1.2 domains linked to a CIDRα1.5 domain to form DC15 (Fig 12A). The NTS-DBLα1.2 domains occurring outside of DC15 are mostly linked to other CIDRα1 domain subclasses (Fig 12B).

## Discussion

Here we have identified four novel rosette-mediating PfEMP1 variants, with evidence from a range of experiments including *var* gene profiling, infected erythrocyte surface expression of

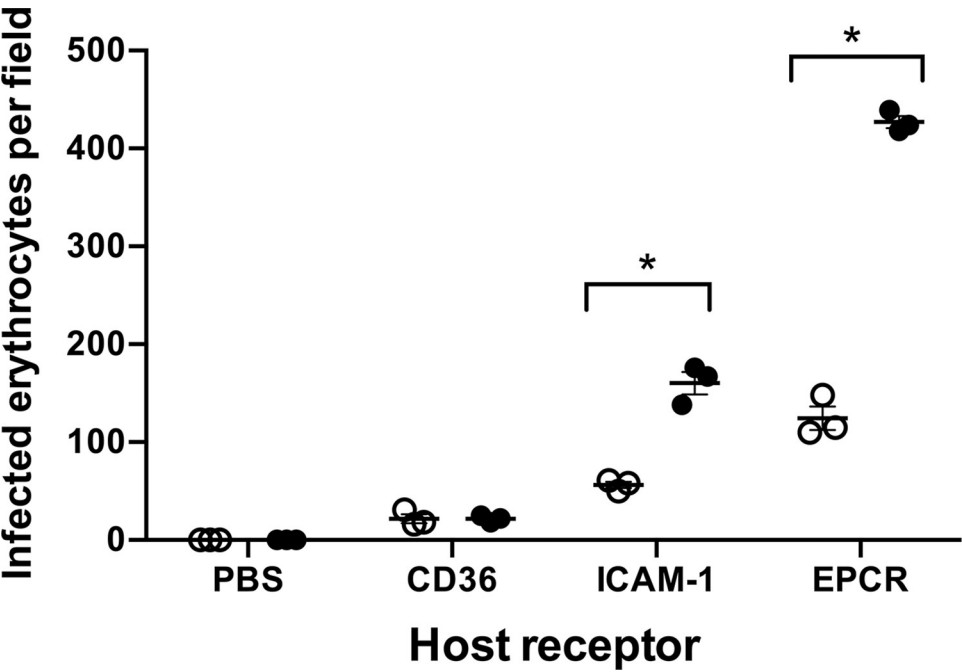

**Fig 10. Spot binding assay of KE08R+ to endothelial cell receptor proteins.** Adhesion of KE08R+ infected erythrocytes to recombinant host receptor proteins was determined with (black circles) and without (white circles) prior rosette disruption with antibodies to KE08VAR_R1 NTS-DBLα. The number of infected erythrocytes bound per field was determined using an inverted microscope with a 40X objective and six fields were counted per spot in each experiment. The difference in mean binding values compared with the same receptor in the presence and absence of rosette from n = 3 independent experiments was analyzed by two-tailed paired t tests corrected for multiple comparisons with the Holm-Sidak method. *P< 0.05.

PfEMP1, rosette disruption with antibodies to PfEMP1 and binding of NTS-DBLα protein to uninfected erythrocytes. Two of the variants identified, KE10VAR_R1 and KE08VAR_R1 contain DC11, giving them comparable PfEMP1 architecture to previously described rosetting variants (Fig 1) [15,17,37]. The two other new rosetting variants, PC0053VAR_R1 and KE11VAR_R1, have a DC15 head structure, not previously implicated in rosetting. Instead, DC15 PfEMP1 variants were thought to bind EPCR via their CIDRα1.5 domain [31]. However, CIDRα1.5 domains fall into two clear subtypes, CIDRα1.5a and CIDRα1.5b, with only the former showing strong EPCR binding [31]. The DC15 variants identified here have an NTS-DBLα1.2 domain that binds erythrocytes (Fig 7) followed by a CIDRα1.5b domain. Thus, rosetting DC15 variants may represent a functionally distinct group characterised by CIDRα1.5b (S5 Fig), although further work is needed confirm this. Our study also showed that the NTS-DBLα2 domain of a "DC8-like" variant encoded by *tm180var1* binds to uninfected erythrocytes, confirming this as a rosette-mediating variant [17].

These findings combined with previous work show that at least four different group A and B/A PfEMP1 types mediate rosetting: DC11, DC16, DC15 (with CIDRα1.5b) and DC8-like, with only the first two containing the recognised rosetting-associated head structure of DBLα1.5/6/8-CIDRβ/γ/δ. It remains unknown whether these different rosette-mediating PfEMP1 types have functional significance, for example, binding to different host receptors on uninfected erythrocytes [58]. Recent work also suggests that a group C PfEMP1 variant PF3D7_0412900 mediates rosetting by binding to Glycophorin B on uninfected erythrocytes [59]. Taken together, these data show that a more diverse set of PfEMP1 types mediates rosetting than originally recognised. Further investigation will be needed to determine whether all

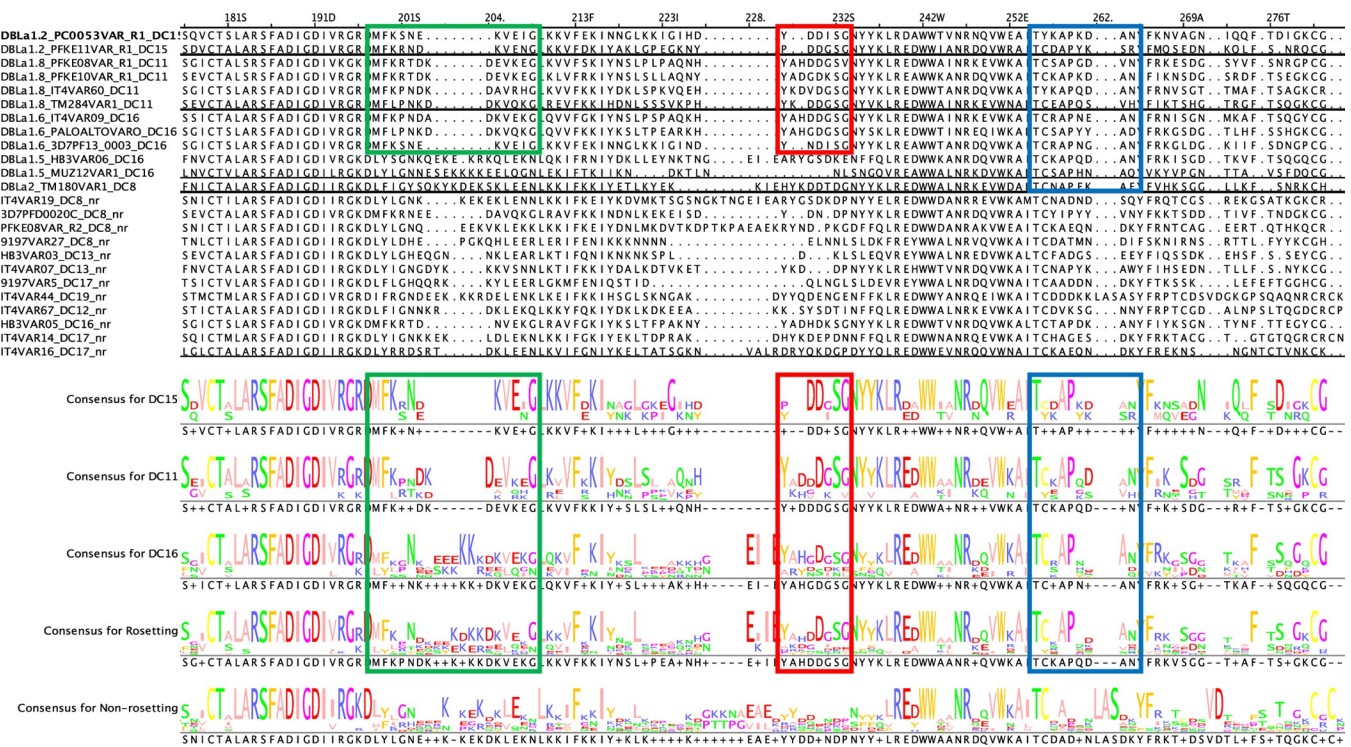

**Fig 11. Rosetting motifs.** The NTS-DBLα amino acid sequences from known rosetting (upper) and non-rosetting (lower) sequences were aligned in Jalview and amino acid motifs over-represented in the rosetting variants are shown in green, red and blue boxes. The amino acid sequences are given in S1 Text. nr: non-rosetting.

PfEMP1 variants containing these particular rosetting-associated DCs mediate rosetting. Our analysis of the *var* gene repertoires of the thirteen Kenyan parasite lines showed that on average, there are three *var* genes per *P. falciparum* genome encoding a DC11 and/or DC16 variant (Table 1). Many of these parasite lines also have one or more *var* genes encoding a DC15 (CIDRα1.5b) and/or a DC8-like variant containing a CIDRα1.8 domain rather than the CIDRα1.1 found in classical non-rosetting EPCR-binding DC8 variants [31]. If the DC types identified here always mediate rosetting, this would imply that about one third of the group A and B/A *var* genes per *P. falciparum* isolate encode rosetting variants, consistent with an important biological role for this phenotype. Exactly how rosetting benefits the parasite is unclear, but a primate model showed that rosetting parasites reach a high parasite burden *in vivo* more rapidly than isogenic non-rosetting parasites [60], which could increase parasite fitness by enhancing transmission [61]. More rapid expansion *in vivo* could be due to rosetting enhancing erythrocyte invasion or reducing immune-mediated clearance, with current experimental evidence favouring the latter hypothesis [62–64].

Previous work has demonstrated a strong association between rosetting and severe malaria [2–5], with the IgM-binding rosetting phenotype being particularly common in parasites collected from African children with severe malaria [40,41]. Whether all rosetting phenotypes mediated by the different DC types shown here have associations with severe malaria is unknown. Analyses of parasite *var* gene transcription and clinical malaria severity have repeatedly shown that expression of group A *var* genes and those encoding DC8 are associated with severe disease [21–25]. As shown here and previously [17,37], the IgM-binding rosetting phenotype is mediated by DC11 or DC16 PfEMP1 variants. Recent studies found DC11 expression was associated with severe malaria in Mozambiquan children [55] and Papuan adults

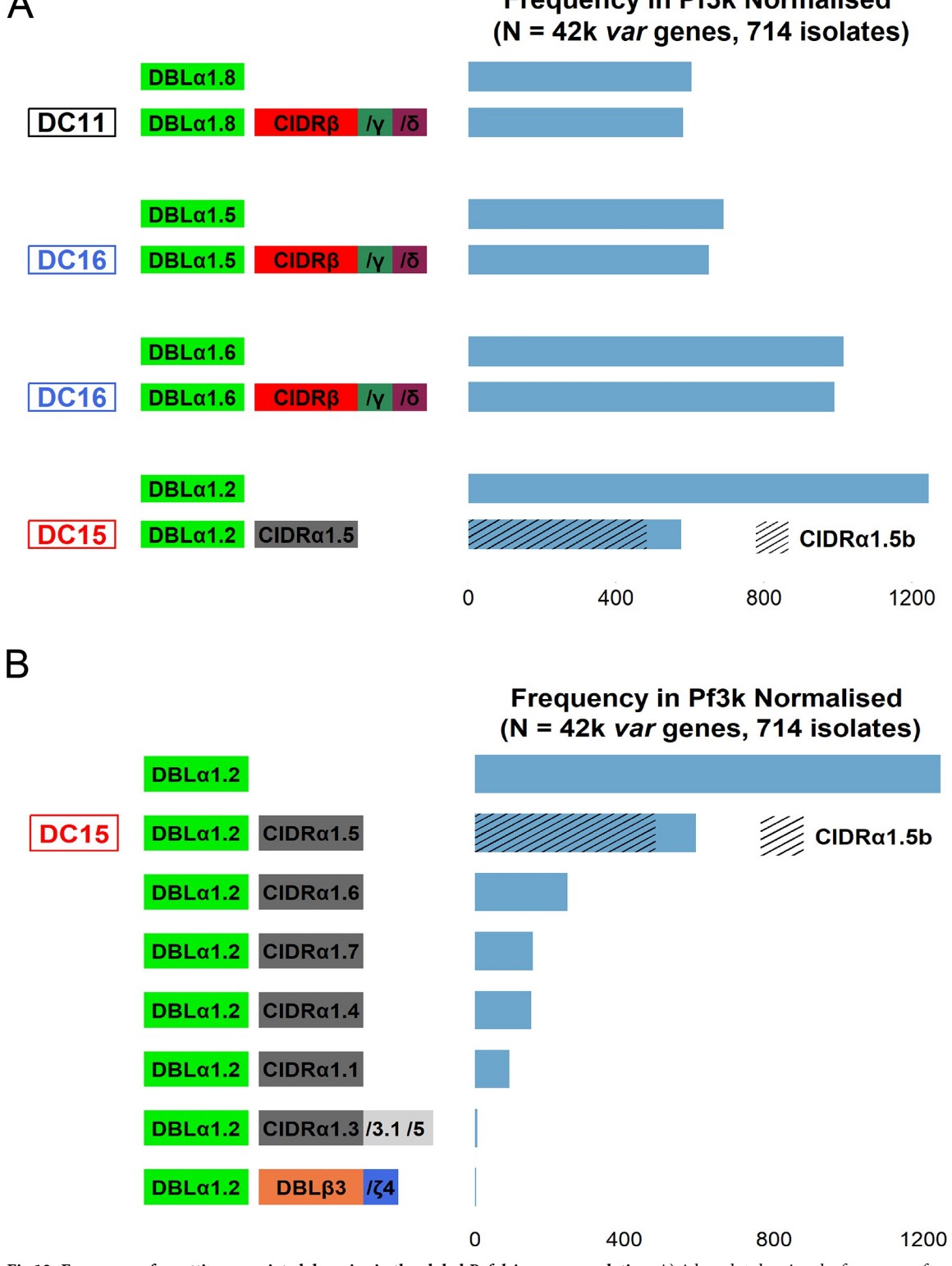

**Fig 12. Frequency of rosetting-associated domains in the global *P. falciparum* population.** A) A bar plot showing the frequency of rosetting-associated domains and their respective domain cassettes in the Pf3k normalised varDB [35]. The domains are colour-coded as defined by Rask et al [27]. B) A bar plot showing the frequency of different domains linked to DBLα1.2 in the Pf3k normalised varDB.

[65], while both DC16 and DC15 were associated with severe malaria in malaria-naïve adults [66]. Many prior studies on *var* gene transcription and malaria severity do not include primer sets for DC11 [22,24,25], raising the possibility that the contribution of this IgM-binding rosette-mediating variant type to severe malaria could be greater than currently appreciated. Understanding how *var* gene transcription relates to parasite adhesion phenotype is important in interpretation of *var* gene profiling/disease severity studies and can help identify the key adhesion phenotypes that could be targeted therapeutically to treat or prevent severe disease.

One of the most underexplored topics in rosetting research is understanding how rosettes sequester *in vivo*. Rosettes do not circulate in the peripheral blood and "rosette-like" aggregates of infected and uninfected cells can be seen in the microvasculature in autopsy studies of cerebral malaria patients [67–69]. Sequestration of rosetting infected erythrocytes may require a dual binding phenotype with simultaneous binding to receptor(s) on microvascular endothelium as well as on erythrocytes [70,71]. However, host endothelial cell receptors for rosetting PfEMP1 variants remain poorly characterised. One in vitro study found that the rosetting line IT4VAR09/R29R+, which expresses a DC16-like variant (Fig 1), adhered to a human brain endothelial cell line via the DBLγ15 domain, which interacted with an unidentified host heparan sulfate proteoglycan [71]. It is notable that all of the experimentally-proven rosetting variants, including the four new ones described here, contain DBLγ as the third or fourth extracellular domain from the N-terminus, and we hypothesize that binding of DBLγ to endothelial cell heparan sulfate is a common mode of sequestration for rosetting infected erythrocytes. The sensitivity of the Kenyan lines to rosette disruption by heparin (a highly sulfated form of heparan sulfate) is consistent with this idea. It is also plausible that the diverse CIDRβ/γ/δ/α1 domains of rosetting variants may bind to additional host receptors, and this remains to be explored.

The ability of some rosette-mediating variants to bind serum proteins such human IgM-Fc and α2M is a phenomenon that clusters PfEMP1 to increase binding avidity [37,38,39,42]. The IgM-binding rosetting variants identified to date all contain a C-terminal triplet of DBLε and DBLζ domains [17], but only one of these domains in each variant binds IgM [37–39,54]. We show here that KE10R+ infected erythrocytes bind IgM via the DBLζ3 domain of KE10VAR_R1, consistent with previous data. Interestingly, the KE08R+ and PC0053R+ parasite lines did not bind IgM, despite their PfEMP1 variants containing a C-terminal DBLε/ζ domain triplet (Fig 4). Therefore, the presence of a C-terminal DBLε/ζ domain triplet is not always predictive of IgM binding, although in the case of KE08R+, the expression of two different PfEMP1 variants at the surface of infected erythrocytes might interfere with binding at the membrane-proximal part of PfEMP1.

The majority of evidence to date suggests that expression of PfEMP1 is mono-allelic [51]. However, here we present evidence suggesting the simultaneous expression of two different group A PfEMP1 variants on the surface of infected erythrocytes of parasite line KE08R+, one mediating rosetting and the other binding to EPCR and ICAM-1. Dual staining with antibodies to the two variants conjugated to different fluorophores was not done, and this would be a useful experiment for future work. A similar finding was reported previously, with a *P. falciparum* 3D7 parasite line expressing two PfEMP1 variants PFD1235w and PF11_0008, conferring a dual binding phenotype to PECAM-1 and ICAM-1, which increased binding efficiency to endothelial cells [52]. Dual binding phenotypes are more commonly described in the context of single PfEMP1 variants, with different domains within the same molecule binding different receptors [30,71]. It seems likely that the ability to adhere to multiple receptors is advantageous to the parasite by strengthening adhesion and thereby promoting sequestration. The multiple adhesion phenotypes seen in both KE08R+ and the 3D7 line described above could be artefactual, for example with loss of mono-allelic *var* gene transcription due to mutations introduced

by the culture adaptation process. Or they could represent genuine aspects of parasite biology that remain to be explored. Existing bulk RNA *var* gene profiling studies in clinical isolates would not detect polygenic *var* gene transcription, and single cell studies of clinical isolates will be needed to determine whether multiple *var* genes are transcribed per infected erythrocyte in natural infections.

The increasing availability of *var* gene sequence data has revolutionised the study of PfEMP1 diversity across multiple *P. falciparum* isolates [35]. However, making functional sense of these data relies on the ability to infer binding phenotype from *var* gene sequence. Excellent progress has been made with PfEMP1 variants that bind to CD36, ICAM-1 and EPCR [29–31], and the work described here is a step towards a more comprehensive understanding of rosette-mediating PfEMP1 variants. Important unanswered questions include how many different PfEMP1 types mediate rosetting, which host receptor(s) are they binding and are they suitable targets for interventions such as strain-transcending antibodies to treat or prevent severe malaria [17,72]?

## Materials and methods

### Ethics statement

Human erythrocytes and serum were obtained after informed written consent from donors at the Scottish National Blood Transfusion Service (SNBTS), Edinburgh, UK. Ethical approval was granted from the University of Edinburgh, School of Biological Sciences Ethical Review Panel (arowe002) and the Scottish National Blood Transfusion Service (SNBTS) Ethics Review Board (19~6 and 22~11).

### Parasite culture and rosette selection

The Kenyan parasite lines (Table 1) were collected from children with severe or uncomplicated *P. falciparum* malaria between 2009 and 2010 in Kilifi and adapted to culture at the KEMRI--Wellcome laboratories [33]. The lines are available from the European Malaria Reagent Repository (http://www.malariaresearch.eu/). Parasites were thawed and cultured in complete RPMI with 5% pooled human serum and 0.25% albumax and blood group O erythrocytes as described [72]. Isogenic pairs of high and low rosetting lines were selected using density gradient methods [73].

### Visualisation of the Kenyan parasite line var gene repertoires

The varDB was accessed from the Wellcome Sanger Institute FTP site (ftp://ftp.sanger.ac.uk/pub/project/pathogens/Plasmodium/falciparum/PF3K/varDB/) using the WinSCP (v5.15.2) FTP client. The PfEMP1 architecture and domain boundaries for the Kenyan parasite lines were obtained from a previous study [35]. The PfEMP1 repertoire for each parasite line was plotted using R v4.2.2 [74] and arranged based on the 5' ups group of each protein (A, B then C). The domains were colour-coded as previously described [27].

### Assessment of rosette frequency

Rosette frequency was assessed by staining an aliquot of culture suspension at ~2% haematocrit with 25 µg/ml ethidium bromide and viewing a wet preparation using simultaneous fluorescence/bright field microscopy on a Leica DM 2000 fluorescent microscope with a x40 objective. A rosette was defined as an infected erythrocyte binding at least two uninfected erythrocytes, and the rosette frequency expressed as the proportion of infected erythrocytes

forming rosettes out of 200 infected erythrocytes counted. Sample identity was masked in rosetting experiments to prevent observer bias.

## IgM-binding flow cytometry

IgM-staining was performed by incubating aliquots of live parasite culture in PBS/1%BSA with 20μg/ml rabbit IgG against PfEMP1 NTS-DBLα for 45 mins then washing three times with PBS. The washed cells were incubated in PBS/1% BSA containing 1/2500 of Vybrant Dye-Cycle stain with 1/1000 Alexa Fluor 647-conjugated cross-absorbed goat anti-rabbit IgG (Invitrogen, A-21244) and 1/1000 Alexa Fluor 488-conjugated cross-adsorbed goat anti-human IgM (heavy chain specific) (Invitrogen, A-21215) before washing as above and fixing in PBS/ 0.5% paraformaldehyde. The samples were acquired on a LSRFortessa instrument (BD Biosciences) with data from least 500 mature infected erythrocytes collected per sample. Data were analysed in FlowJo v.10 using the gating strategy shown in Fig 2A. As a negative control, parasites were grown as described above in complete RPMI but with 5% IgM-deficient human serum, prepared by five consecutive 30 minute incubations with anti-human IgM beads (Sigma-Aldrich A9935).

## Trypsinisation of parasitised erythrocytes

Mature pigmented trophozoites were purified from 100μl packed cell volume of parasite culture at >4% parasitaemia using magnetic purification as described [72]. The purified parasites were washed twice then resuspended in 100μl of trypsin (Sigma-Aldrich T4799) in PBS at concentrations 0, 0.5, 1 and 5μg/ml, and incubated at 37°C for 5 mins. Cells were pelleted by centrifugation and the pellet resuspended in 1 mg/ml soybean trypsin inhibitor (Sigma-Aldrich T9003) and incubated at 37°C for 5 mins. The cell pellet was washed and resuspended in complete binding media (RPMI 1640 without bicarbonate with 25 mM HEPES, 16mM glucose, 25μg/ml gentamicin, and 10% human serum). O+ erythrocytes were added to give a final parasitaemia of 20%, mixed, and incubated at 37°C for 1 hour. The cells were then pelleted before being resuspended by gentle pipetting and rosette frequency assessed. Samples were assayed in duplicate and the mean of the two rosette frequency counts taken.

## Rosette disruption by heparin

Parasite cultures in complete binding medium were incubated with 1 mg/ml or 0.1 mg/ml heparin (Sigma H4784) for 30 mins at 37°C and rosetting was assessed as described above and compared to a control with no added heparin.

## DBLα sequence tag RT-PCR

RNA was extracted from parasite cultures at the mid-late ring stage (~10–20 hours post invasion) by resuspending pelleted cells in 10x volumes of Trizol (Ambion 15596026). RNA was extracted using bromochloropropane (Sigma-Aldrich B9673) and a Zymo Research RNA Clean & Concentrator Kit (R1014) according to the manufacturer's instructions. Samples were treated with TURBO DNase (Invitrogen AM1907) and cDNA made with SuperScript III First-Strand Synthesis System (Invitrogen 18080–051) with random hexamers. DBLα-tag 50 μl PCR reactions contained 2μl of cDNA, 20.75 μl of dH$_2$O, 4 μl of 25mM MgCl$_2$, 8 μl of 1.25mM dNTPs (Promega U120A, U121A, U122A, U123A), 5 μl of αAF2 primer (GCACG(A/C) AGTTT(C/T)GC) [47], 5 μl of αBR primer (GCCCATTC(G/C)TCGAACCA) [47], and 5 μl of Amplitaq Gold buffer with 0.25 μl Amplitaq Gold (1.25 units) (Applied Biosystems 4311816). The reactions were cycled as follows, 95°C for 5 mins, 95°C for 20 s, 42°C for 20 s, 60°C for 1

minute, 60°C for 10 mins, 4°C to hold. The PCR products were cleaned using a Nucleospin Gel and PCR Clean-up kit (Macherey-Nagel 740609.250) then cloned using a TA Cloning Kit Dual Promoter pCRII Vector (Invitrogen K207020), at a vector to insert ration of 1:3. TOP10F' One Shot Competent Cells (Invitrogen 440300) were transformed using 2μl of each ligation reaction. 24 colonies were picked per ligation and grown overnight before plasmids were isolated from the cultures using a GeneJET Plasmid Miniprep Kit (ThermoFisher Scientific K0502). Plasmid inserts were Sanger sequenced at the MRC Protein Phosphorylation and Ubiquitiylation Unit, Dundee (https://dnaseq.co.uk/) using the forward primer VF1 (AGCTCGGATCCACTAGTA) and the resulting amino-acid sequences were used in a BLAST search [75] against the *var* gene/PfEMP1 repertoire of the relevant *P. falciparum* line to identify the full-length gene sequence.

### *pfke10var_r1* sequence extension

The *pfke10var_r1* Exon 1 sequence was incomplete in the database, so a forward primer (11019F2 5' GATTTTGACACATC CAAAGTGT) was designed targeting the *pfke10var_r1* DBLζ3 domain ~200bp from the end of the partial sequence. This was paired with a reverse primer targeting a conserved region in Exon 2 (Exon2_3.3 5' CCATTTCTTCATATT-CACTTTC) [76]. 50pmols of each primer was added to a reaction mix containing 1.25mM dNTPs, 1.5mM MgCl$_2$, 1x Platinum *Pfx* buffer (Invitrogen 11708), 1 unit of Platinum *Pfx* DNA polymerase (Invitrogen 11708), 0.2μg cDNA and made up to a final reaction volume of 50μl. The PCR conditions were initial denaturation at 95°C for 5 mins followed by 35 cycles of denaturation at 95°C for 5 secs, annealing at 52°C for 15 secs and extension at 60°C for 3 mins. A final extension was performed at 60°C for 10 minutes and the amplicon was resolved in a 1% agarose gel at 70V for 50 minutes to reveal a 2kb band. This was gel purified using a purification kit (Macherey-Nagel 740609) following the manufacturer's instructions. The purified PCR product was Sanger sequenced using both the forward and reverse sequences listed above. From the newly generated sequence from the forward primer, another primer sequence was designed (11019P1 5' GAGTGTAAAGTTGAGTCCCTTG) targeting a region ~400bp downstream which was then used to further sequence the original 2kb PCR product. The Sanger sequencing results from the three primers (11019F2, 11019P1 and Exon 2) and the partial Exon 1 sequence were then aligned to generate the complete Exon 1 sequence. The Var-Dom server [27] was used to define the domain boundaries of the complete KE10VAR_R1 variant (https://services.healthtech.dtu.dk/services/VarDom-1.0/). The subclasses were resolved using the individual domains in a BLAST search [75] against the Pf3k database [35] and identifying the subclass of the top hit based on full length alignment.

### Recombinant protein production

Codon optimised sequences for *E. coli* expression of PfEMP1 domains with a Bam HI site at the 5' end and an Nhe I site at the 3' end were obtained from Twist Biosciences (San Francisco, California) and ligated into the expression vector modified pET15b (pET15b conv) [77]. The DBLα domain sequences included the N-terminal sequence and an N-terminal His-tag. Ligation reactions were sequenced to confirm the correct insert sequence. Origami B cells containing pRIG were transformed and protein expression was induced by the addition of 1mM IPTG and the bacteria harvested. Cell pellets were frozen before resuspension in Bugbuster mastermix (Novagen 71456) and EDTA-free protease inhibitor tablets (Roche 11836170001). Nickle-NTA or cobalt-NTA chromatography was used to purify the His-tagged proteins followed by gel filtration as described [17]. For protein KE10VAR_R1 DBLα and TM180VAR1 DBLα the His-tag was cleaved with TEV protease before use in experiments [17,49]. Fractions

containing protein were pooled and buffer exchanged into PBS using PD-10 desalting columns (Sigma-Aldrich GE17-0851-01). The proteins were concentrated using Vivaspin 20ml 10 kDa or 30 KDa MWCO concentrators (Merck Z614602-12EA) and stored in aliquots at -70°C.

## Rabbit immunisations and IgG purification

Rabbit immunisations were done by BioGenes GmBH (Berlin, Germany). Preimmune sera were screened to identify rabbits that did not have anti-human erythrocyte antibodies by immunofluorescence assay as described [17,49]. Selected rabbits were immunised with 250μg of antigen in BioGenes proprietary adjuvant (92.8% mineral oil, 3.48% Tween-20, 3.48% Span-80 and 0.23% lipopolysaccharides from *Phormidium spp*) on day 1 followed by boosting with 100μg of antigen on days 7, 14, 28, 49 with the final bleed on day 56. Total IgG was purified from the resulting antisera using a Pierce protein A IgG purification kit (ThermoFisher Scientific 44667) according to the manufacturer's instructions.

## Flow cytometry with PfEMP1 antibodies

Staining of mature parasitised erythrocytes for PfEMP1 with rabbit IgG was done as described [72]. Briefly, whole culture was incubated for 45 minutes with rabbit IgG against the relevant PfEMP1 domain at a final concentration of 20μg/ml before being washed three times with PBS. The cells were then incubated in PBS/1% BSA containing 1/2500 of Vybrant DyeCycle stain with 20μg/ml ethidium bromide and 1/1000 Alexa Fluor 647-conjugated cross-adsorbed goat anti-rabbit IgG (Invitrogen A-21244) for 30 minutes. Cells were washed twice in PBS and once in PBS 1% BSA before being fixed in PBS/0.5% paraformaldehyde. The samples were acquired on a LSRFortessa instrument (BD Biosciences) in duplicate with at data from least 500 mature parasitised erythrocytes collected per sample. Data were analysed with FlowJo v.10. For trypsinisation experiments, cultures were washed twice in incomplete binding media (RPMI 1640 without bicarbonate with 25 mM HEPES, 16mM glucose and 25μg/ml gentamicin) before being resuspended at 5% haematocrit in 5μg/ml trypsin (Sigma-Aldrich T4799) in incomplete binding medium and incubated at 37°C for 5 mins. Cells were washed twice in complete binding medium (as above plus 10% human serum) and stained as described above.

## Rosette disruption assays

Parasite culture at ~2% haematocrit was pre-stained with 25μg/ml ethidium bromide for 2 mins at 37°C. The cells were pelleted by centrifugation, the supernatant removed, and the pellet resuspend at 2% haematocrit in complete binding medium as described above. The relevant anti-PfEMP1 rabbit IgG was added to the stained culture to give final concentrations of 100, 10, 1 and 0.1μg/ml. An equivalent volume of PBS was added to the no antibody control. The samples were incubated at 37°C for 1 hour with gentle resuspension of cells every 10 mins before assessment of rosette frequency. Statistical analyses and graphs were done using GraphPad Prism (Version 10.2.2, GraphPad software, Inc).

## Erythrocyte binding assays

The assay investigating the ability of recombinant PfEMP1 domains to bind to uninfected erythrocytes was based on those described previously [53,78]. 10 μl of washed O+ erythrocytes were re-suspended in 1ml PBS/1% BSA with 100μg/ml of recombinant protein and incubated on ice for 1 hour then washed twice with 200 μl ice-cold PBS and resuspended in 23.75 μl of PBS/1% BSA. 1.25 μl of the relevant rabbit IgG to the recombinant protein PfEMP1 domain at 1 mg/ml (final concentration 50 μg/ml) was added and incubated on ice for 45 mins before

washing twice with 500μl PBS. The washed cell pellets were resuspended in 50μl of PBS/1% BSA containing 1:1000 dilution of Alexa Fluor 488 goat anti-rabbit IgG (Invitrogen A-11034) and incubated for a further 45 mins on ice in the dark before being washed twice. The cells were resuspended in 200μl of PBS containing 0.1% BSA and 0.01% sodium azide and transferred to a 96 well plate. The 96 well plates were read on a LSRFortessa instrument (BD Biosciences). Data were analysed in FlowJo v.10. For statistical analysis, median fluorescence intensity (MFI) values were log transformed and analysed by one way ANOVA in GraphPad Prism v10.

## IgM Enzyme Linked Immunosorbent Assay (ELISA)

An ELISA was used to test the interaction between IgM and KE10VAR_R1 domains as previously described [37], with modifications. ELISA plates (Greiner 655061) were coated with 200 μl of 10 μg/ml human IgM (Sigma 18260) diluted in 0.1 M carbonate-bicarbonate buffer and incubated at 4°C overnight. Control wells contained buffer but no IgM. After incubation, the coating buffer was flicked off and blocking buffer (PBS/5% skimmed milk/0.1% (v/v) Tween 20) was added for 2 hours at RT. The plate was washed three times with PBS. His-tagged KE10VAR_R1 DBL proteins were suspended in dilution buffer (PBS/1% skimmed milk/0.1% (v/v) Tween 20) and 150 μl of each DBL protein at four 5-fold dilutions (50 μg/ml, 10 μg/ml, 2 μg/ml, 0.4 μg/ml) were added to duplicate wells and incubated overnight at 4°C. After incubation, the plate was washed three times with PBS. 50 μl of anti-His HRP (1:3000) (Biolegend 652404) in dilution buffer was added to each well and incubated at 37°C for 1 hour. The plate was washed three times with PBS and twice with deionised water. 50 μl of 3,3′,5,5′-Tetramethylbenzidine (TMB) (Sigma, T2885) substrate was added to each well and incubated at RT for 7 mins. The reaction was stopped using 0.18 M sulphuric acid and absorbance read at 450nm. The mean absorbance readings from duplicate wells were plotted against DBL protein concentration using R v4.2.2 [74].

## Host receptor spot binding assays

3 μl spots of recombinant ICAM-1 (50 μg/ml; R&D 720-IC), EPCR (50 μg/ml; Sino Biologicals 13320-H02H), CD36 (25 μg/ml; R&D 1955-CD) and PBS, all in duplicate were spotted onto 60 mm diameter bacteriological petri dishes (BD Falcon 351005) and incubated in a humidified chamber overnight at 4°C. Next day the spots were aspirated and 2 ml of PBS/2% BSA blocking buffer was added per dish and incubated for 2 hours at 37°C. During the blocking step, 1.5 ml aliquots of KE08R+ culture suspension at 5–10% parasitemia (predominantly mature pigmented trophozoite stage) and ~2% Ht were prepared by pre-incubation with or without 10 μg/ml of rabbit polyclonal IgG to KE08VAR_R1 NTS-DBLα for 30 mins at 37°C to disrupt rosettes. The blocking solution was removed, dishes washed with incomplete binding medium (as described above but without serum) and 1.5 ml of the prepared culture suspension was added to each dish. Dishes were incubated at 37°C for 1 hour, with gentle resuspension every 12 minutes. Unbound cells were removed by at least five gentle washes with 4 ml of incomplete binding medium per wash; bound cells were fixed with PBS/2% glutaraldehyde for at least 30 mins and stained with 5% Giemsa for 10 mins. The number of infected erythrocytes bound to each spot was counted with an inverted microscope using a 40X objective. Technical replicates based on counts of duplicate spots or dishes were averaged to determine a binding value for each receptor in each experiment, and statistical analysis was done on the average binding values from $n \geq 3$ independent experiments using one-way ANOVA with Dunnett's multiple comparisons test or two-tailed paired t tests with Holm-Sidak multiple comparisons test.

**Table 2.** *var* genes encoding rosetting PfEMP1 variants*.

| S/N | *var* gene/alternative name | Group** | DBLα class | DC | Ref. |
|---|---|---|---|---|---|
| 1 | pc0053var_r1/9197var15 | A | 1.2 | 15 | This study |
| 2 | pfke11var_r1/pfke11.g448 | A | 1.2 | 15 | This study |
| 3 | pfke08var_r1/9605.g502 | A | 1.8 | 11 | This study |
| 4 | pfke10var_r1/11019var1 | A | 1.8 | 11 | This study |
| 5 | IT4var60 | A | 1.8 | 11 | [15] |
| 6 | TM284var1 | A | 1.8 | 11 | [17] |
| 7 | IT4var09 | A | 1.6 | 16 | [14] |
| 8 | PaloAltovarO | A | 1.6 | 16 | [32] |
| 9 | 3D7PF13_0003 | A | 1.5 | 16 | [16] |
| 10 | HB3var06 | A | 1.5 | 16 | (17) |
| 11 | Muz12var1 | A | 1.5 | 16 | [17] |
| 12 | TM180var1 | B | 2 | 8 | [17] |

*Parasite lines expressing these variants have been examined microscopically and shown to form rosettes and evidence that the PfEMP1 NTS-DBLa binds erythrocytes has been provided.

** Ups sequence from Rask et al [27] or Otto et al [35].

S/N: sequence number in the alignment; DC: domain cassette; Ref: reference.

Statistical analyses and graphs were done using GraphPad Prism (Version 10.2.2, GraphPad software, Inc).

## Multiple sequence alignment and motif search

Multiple sequence alignment (MSA) of the NTS-DBLα protein sequences were performed using Clustal Omega, accessible through the European Bioinformatics Institute website (https://www.ebi.ac.uk/Tools/msa/) using default parameters. The MSA output was analysed using Jalview [79] to assess the presence of known rosetting motifs across rosetting and non-rosetting variants. The sequence analysis included NTS-DBLα sequences from experimentally validated PfEMP1 variants, comprising 12 rosetting (Table 2) and 12 non-rosetting (Table 3) variants.

**Table 3.** *var* genes encode non-rosetting PfEMP1 variants*.

| S/N | *var* gene | Group** | DC | Associated phenotype | Ref. |
|---|---|---|---|---|---|
| 1 | IT4var19 | B | 8 | HBEC-binding*** | [50] |
| 2 | 3D7PFD0020c | A | 8 | HBEC-binding | [50] |
| 3 | pfke08var_r2/PFKE08.g501 | B | 8 | ICAM-1, EPCR | this study |
| 5 | HB3var03 | A | 13 | HBEC-binding | [50] |
| 6 | IT4var07 | A | 13 | HBEC-binding | [50] |
| 7 | 9197var5 | B | 17 | HBEC-binding | S1 Text |
| 8 | IT4var44 | B | 12 | Clumping | S1 Text |
| 9 | IT4var67 | B | 12 | Clumping | S1 Text |
| 10 | HB3var05 | A | 16 | Unknown | S1 Text |
| 11 | IT4var14 | B | 14 | CD36- and ICAM-1 binding | [80] |
| 12 | IT4var16 | B | 14 | CD36- and ICAM-1 binding | [81] |

*Parasite lines expressing these variants have been examined microscopically and no rosettes were seen.

** Ups sequence from Rask et al [27] or Otto et al [35].

*** HBEC, human brain endothelial cell

S/N: sequence number in the alignment; DC: domain cassette; Ref: reference.

## Frequency of rosetting domains in the Pf3k database

The frequency of individual rosette-mediating domains (DBLα1.8, DBLα1.5, DBLα1.6, DBLα1.2) and DCs (DC11, DBLα1.8-CIDRβ/γ/δ; DC16, DBLα1.5-CIDRβ/γ/δ or DBLα1.6--CIDRβ/γ/δ and DC15, DBLα1.2-CIDRα1.5) were determined from the subclass-annotated Pf3k normalised varDB dataset [35] using Linux (Ubuntu v20.04.5 LTS). The total number of each individual domain class was assessed, and the instances in which each DBLα domain was followed by the respective DC domain(s) along the architecture were established. For further analysis of DBLα1.2, the frequency of each unique head structure associated with DBLα1.2 was established by extracting the first two domains of each DBLα1.2 instance. To determine the frequency of DC15 (DBLα1.2-CIDRα1.5b) specifically, all the DBLα1.2-CIDRα1.5 domains were extracted from the database and their amino acid sequences aligned using Clustal O (v1.2.4) [82]. IQTREE (v2.0.5) [83] was used to generate a maximum likelihood tree (1000 bootstraps). The CIDRα1.5b clade comprised all sequences in the same clade as PC0053-C.g687 (PC0053VAR_R1) and PFKE11.g448 (PFKE11VAR_R1) based on ≥80% bootstrap support. The results were plotted on a bar plot in R v4.2.2 [74,84].

## CIDRα1.5 analysis of the Kenyan parasite lines

The amino acid sequences of CIDRα1.5 domains from the Kenyan parasite lines were extracted based on previously published domain boundaries [35]. The sequences were aligned with default parameters using MUSCLE (v3.8.1551) [85] and IQTREE (v2.0.5) was used to generate a maximum likelihood tree [83]. The IQTREE command "*iqtree2—safe -s cidra.aa. aligned.fasta -m JTT+F+I+I+R10 -alrt 1000*" was used and the results were visualised and annotated using ggtree in R [74,84].

## Supporting information

**S1 Fig. PfEMP1 repertoires of the Kenyan parasite lines.**
(PDF)

**S2 Fig. SDS-PAGE of recombinant PfEMP1 domains.** SDS-PAGE of recombinant PfEMP1 domains. SDS-PAGE images of the recombinant PfEMP1 domains used to generate antibodies and/or used in erythrocyte binding assays and ELISAs. The his-tagged proteins were expressed in E. coli and purified by Ni-NTA or Co-NTA affinity chromatography, followed by size exclusion chromatography in some cases (this was done for all of the DBLα proteins except KE11VAR_R1 and PC0053-C.g410). Proteins IT4VAR60 DBLα, KE10VAR_R1 DBLα, PC0053VAR_R1 DBLα and PC0053-C.g96 DBLα had the his-tag removed by TEV cleavage as described previously (17, 48), whereas the other proteins did not. Gels were 4–12% Bis-Tris Novex gels run with MOPS or MES buffer at 200V for 55 minutes and stained with Instant Blue. Domain names are given below each image, except for KE10VAR_R1 domains which are lanes 1) Benchmark ladder 2) PFKE10VAR_R1 DBLα1.8 3) PFKE10VAR_R1 DBLγ7 4) PFKE10VAR_R1 DBLε11 5) PFKE10VAR_R1 DBLζ3 and 6) PFKE10VAR_R1 DBLε8. Molecular weight markers in kDa are shown for each gel. The NTS-DBLα preparations gave >90% of protein at the expected molecular weight (~50 kDa), whereas the other domain types had some degraded fragments and/or dimers.
(DOCX)

**S3 Fig. Gating strategy to identify PfEMP1 on mature infected erythrocytes.** A) Forward and side scatter were used to gate on erythrocytes (RBC) and exclude debris. B) Mature pigmented-trophozoite- and schizont-infected erythrocytes (Mature IEs) were detected as the DNA/RNA high population by staining with 1/2500 dilution of Vybrant DyeCycle Violet

(DNA stain) and 20μg/ml of ethidium bromide (DNA/RNA stain). C) PfEMP1 on the surface of live mature infected erythrocytes was detected with 20μg/ml of polyclonal rabbit IgG against NTS-DBLα (variant KE08VAR_R1 shown) followed by 1/1000 dilution of Alexa Fluor 647-conjugated goat anti-rabbit IgG secondary antibody (red). The negative control (blue) was the same parasite culture suspension stained with rabbit IgG against NTS-DBLα from an irrelevant PfEMP1 variant (HB3VAR03 shown) or non-immunised rabbit IgG.
(DOCX)

**S4 Fig. Effect of low dose trypsinisation on PfEMP1 staining.** Parasite cultures were incubated for 5 mins at 37˚C with buffer only "Control" or with 5 μg/ml of trypsin, then stained with 20 μg/ml of PfEMP1 antibody (blue) or negative control rabbit IgG (red). A-B) PFKE08R+ with PFKE08VAR_R2 antibody; C-D) PFKE08R+ with PFKE08VAR_R1 antibody; E-F) PFKE10R+ with PFKE10VAR_R1 antibody. G-H) PC0053R+ with PC0053VAR_R1 antibody. One representative experiment out of two is shown.
(DOCX)

**S5 Fig. Phylogenetic tree of CIDRα1.5 domains from the Kenyan parasite lines.** IQTREE was used to generate a maximum likelihood tree of the CIDRα1.5 amino acid sequences from the PFKE parasite line PfEMP1 repertoires, which had been aligned using MUSCLE. The amino acid domain boundaries were obtained from a previous study (27). Percentage bootstrap support is indicated on the nodes based on 1000 replicates, and the scale bar represents the number of changes per site. The CIDRα1.5b domains from the two rosetting variants are indicated in red (PFKE11.g448 = PFKE11VAR_R1 and PC0053-C.g687 = PC0053VAR_R1) and are found within a distinct clade within the tree supported by high bootstrap values. In this example, the tree is used as a way of visualising similarity between variants and is not intended to infer evolutionary descent.
(DOCX)

**S1 Table. *var* gene profiling of the Kenyan parasite lines.**
(PDF)

**S1 Text. NTS-DBLα amino acid sequences of the variants included in the multiple sequence alignment.**
(TXT)

**S1 Data. Underlying data for figures.**
(CSV)

## Acknowledgments

This work is published with the permission of the Director of KEMRI.

## Author Contributions

**Conceptualization:** J. Alexandra Rowe.

**Data curation:** Florence E. McLean, Brian R. Omondi, Nouhoum Diallo, J. Alexandra Rowe.

**Formal analysis:** Florence E. McLean, Brian R. Omondi, Nouhoum Diallo, Stanley Otoboh, Carolyne Kifude, J. Alexandra Rowe.

**Funding acquisition:** Florence E. McLean, Brian R. Omondi, Rivka Lim, J. Alexandra Rowe.

**Investigation:** Florence E. McLean, Brian R. Omondi, Nouhoum Diallo, Stanley Otoboh, Carolyne Kifude, Rivka Lim, Ashfaq Ghumra, J. Alexandra Rowe.

**Methodology:** Florence E. McLean, Brian R. Omondi, Stanley Otoboh, Carolyne Kifude, Abdirahman I. Abdi, Rivka Lim, Thomas D. Otto, J. Alexandra Rowe.

**Resources:** Abdirahman I. Abdi, Thomas D. Otto, Ashfaq Ghumra.

**Supervision:** J. Alexandra Rowe.

**Validation:** J. Alexandra Rowe.

**Visualization:** Florence E. McLean, Brian R. Omondi, Nouhoum Diallo, J. Alexandra Rowe.

**Writing – original draft:** Florence E. McLean, Brian R. Omondi, J. Alexandra Rowe.

**Writing – review & editing:** Florence E. McLean, Brian R. Omondi, Nouhoum Diallo, Stanley Otoboh, Carolyne Kifude, Abdirahman I. Abdi, Rivka Lim, Thomas D. Otto, Ashfaq Ghumra, J. Alexandra Rowe.

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
