## [Decision Letter · Decision Letter 0]

13 Sep 2024

Dear Professor Rowe,

Thank you very much for submitting your manuscript "Identification of novel PfEMP1 variants containing domain cassettes 11, 15 and 8 that mediate the Plasmodium falciparum virulence-associated rosetting phenotype" for consideration at PLOS Pathogens. As with all papers reviewed by the journal, your manuscript was reviewed by members of the editorial board and by several independent reviewers. The reviewers appreciated the attention to an important topic. Based on the reviews, we are likely to accept this manuscript for publication, providing that you modify the manuscript according to the review recommendations.

This manuscript uses Kenyan field isolates to characterise several novel PfEMP1 sequences that are implicated in resetting. This study identified a new variant of DC cassette involved in resetting, therefore expanding the current knowledge of sequences that mediate rosetting. On the whole the work in the manuscript was solid. The concerns raised by the reviewers that need to be addressed are:

Major concerns

1. Experimental: The authors show by FACS that the KE08R+ line stained positively with two different PfEMP1 antibodies, indicates they are expressing two variants. This should be validated by IFA and show co-staining with two anti-PfEMP1 antibodies. Furthermore, the authors use trypinisation to demonstrate rosetting is mediated by PfEMP1, and FACS or western to demonstrate this loss of PfEMP1 is indeed responsible would validate this.

2. What is the frequency of known rosetting DBLa1.2/1.5/1.6/1.8 domains and the rosetting motifs within or outside cassettes. This information could be gleaned from the abundant var gene assemblies from the Pf3K project provided by Otto et al Wellcome Open Research 2019.

Minor concerns

1. The authors in the manuscript first discuss about field isolates and then role of TM180 var in rosetting. Suggestion is to revise the order and discuss first about TM180 followed by the data on field isolates.

2. Fix references as indicated by Reviewer 3, including providing more current and accurate reviews for the model of monoallelic expression.

3. The manuscript was poorly written in parts and contained many spelling mistakes that detracted from the manuscript. The manuscript needs proof reading to ensure these errors are fixed.

Sincerely,

Tania F. de Koning-Ward

Academic Editor

PLOS Pathogens

Tracey Lamb

Section Editor

PLOS Pathogens

Michael Malim

Editor-in-Chief

PLOS Pathogens

orcid.org/0000-0002-7699-2064

This manuscript uses Kenyan field isolates to characterise several novel PfEMP1 sequences that are implicated in resetting. This study identified a new variant of DC cassette involved in resetting, therefore expanding the current knowledge of sequences that mediate rosetting. On the whole the work in the manuscript was solid. The concerns raised by the reviewers that need to be addressed are:

Major concerns

1. Experimental: The authors show by FACS that the KE08R+ line stained positively with two different PfEMP1 antibodies, indicates they are expressing two variants. This should be validated by IFA and show co-staining with two anti-PfEMP1 antibodies. Furthermore, the authors use trypinisation to demonstrate rosetting is mediated by PfEMP1, and FACS or western to demonstrate this loss of PfEMP1 is indeed responsible would validate this.

2. What is the frequency of known rosetting DBLa1.2/1.5/1.6/1.8 domains and the rosetting motifs within or outside cassettes. This information could be gleaned from the abundant var gene assemblies from the Pf3K project provided by Otto et al Wellcome Open Research 2019.

Minor concerns

1. The authors in the manuscript first discuss about field isolates and then role of TM180 var in rosetting. Suggestion is to revise the order and discuss first about TM180 followed by the data on field isolates.

2. Fix references as indicated by Reviewer 3, including providing more current and accurate reviews for the model of monoallelic expression.

3. The manuscript was poorly written in parts and contained many spelling mistakes that detracted from the manuscript. The manuscript needs proof reading to ensure these errors are fixed.

Reviewer Comments (if any, and for reference):

Reviewer's Responses to Questions

**Part I - Summary**

Reviewer #1: Florence McLean et. al., manuscript has good observations. I am overall pleased with the data in the manuscript. This is very much in the right direction. I would be very pleased to see few additional data which is going to add more value to the manuscript.

Study using field isolates are difficult but very useful for the malaria community as they give the real picture of where the disease is headed. Therefore, it holds a lot of significance in malaria research and our understanding about malaria parasite. Please find my comments below.

Reviewer #2: The research article ‘Identification of novel PfEMP1 variants containing domain cassettes 11, 15 and 8 that mediate the Plasmodium falciparum virulence associated rosetting phenotype’ demonstrates the role of other DC cassettes in rosetting. The authors identified PfEMP1s expressed in kenyan field isolates and studied their role in rosetting. Importantly authors showed a new variant of DC cassette involved in rosetting where they have performed experiments to directly co-relate PfEMP1s with rosetting or cytoadherence phenomena. They also identified PfEMP1s expressed in field isolates and using antibody inhibition assays established their roles in rosetting. Their work also depicted that IgM binding of one of the isolates suggesting this phenotype is found among clinical isolates.

Reviewer #3: The paper identifies and characterises several novel PfEMP1 sequences that are implicated in mediating the pathogenic phenotype of rosetting. This is useful information that will aid the elucidation of the mechanism of rosetting which is a potential therapeutic and vaccine target. The identified sequences were thoroughly characterised and the DBLa1.2/1.8 sequences were shown to bind erythrocytes, consistent with previous evidence that DBLa1.5/1.6/1.8 are responsible for rosetting. The main novelty here seems to lie in extending the classification of rosetting domains to DBLa1.2 and including the DC15 cassette as present in rosetting PfEMP1s.

**Part II – Major Issues: Key Experiments Required for Acceptance**

Reviewer #1: 1. Since Authors have all required antibodies available it would be nice to see co-staining with two anti-PfEMP1 antibodies on the same parasite with KE08R+, where authors have shown expression of two PfEMP1.

2.Authors already have all required antibodies against PfEMP1s. It would be very nice to include a figure showing loss of PfEMP1 upon trypsinization using FACS/or western blot analysis.

Reviewer #2: 1.The authors in the manuscript first discuss about field isolates and then role of TM180 var in rosetting. In my opinion the authors should discuss first about TM180 followed by their data on field isolates.

Although the authors mention that the work on the host receptor is for future studies. However inhibition experiment with heparan sulfate would provide an idea whether the rosetting is similar to other DC cassettes especially for the parasite that shows IgM binding phenotype

Reviewer #3: The conserved domain cassette structure of var genes has featured a lot in attempts to associate sequences with disease and this makes sense as selection for adhesion seems the most probable reason for the conserved domain cassettes. In this light the information on DC15 is useful but could include a succinct exploration of the frequency of known rosetting DBLa1.2/1.5/1.6/1.8 domains and the rosetting motifs within or outside cassettes. There are abundant var gene assemblies from the Pf3K project provided by Otto et al Wellcome Open Research 2019 that could be used for this.

**Part III – Minor Issues: Editorial and Data Presentation Modifications**

Reviewer #1: Manuscript has a lot of sections poorly written and contains errors that should not be part of final draft. I highly encourage authors to go through the manuscript multiple times to remove those mistakes which otherwise restricts the flow for readers. I do understand these are benign mistakes but it is best if they could be minimised.

Reviewer #2: 1. The authors should perform spell check like ‘analyses’ in line 331 and ‘alos’ in line 356.

Reviewer #3: The authors also identify a parasite that expresses two PfEMP1 simultaneously, in addition to the cited previous report of this phenomenon by Joergensen 2010 it has also been reported previously by Brolin 2009 Genome Biol, albeit the co-expressed genes were duplicated copies of var2csa. It is possible that dual expression is of biological relevance, it has however been shown by reverse genetics by Voss et al, as cited in the MS, but also by Dzikowski 2006 PLoS Pathog, that selection of a recombinant var gene silences other var genes. Although these experiments were all performed on the NF54-3D7 lineage, ItG has also been selected for monoallelic expression of A4 var in repeated publications by the Newbold lab, eg Kyes 2007 Mol Micro and for monoallelic expression of var2csa by multiple labs eg Duffy 2005 Mol Micro. I think the extent of experimental support for the model of monoallelic expression could be more clearly communicated, perhaps cite a current and accurate review? Although the cited Scherf paper was the first paper to propose monoallelic expression it proposes a disproven model of promiscuous ring stage expression and exclusive expression in trophozoites based on the expression of var1 that was subsequently shown to be atypical and constitutive in all parasites and is probably independent of allelic exclusive expression. I would recommend replacing this reference with a subsequent reference that accurately explains monoallelic expression as we understand it now, or at least supplementing it with a few such references to avoid readers drawing incorrect conclusions from the Scherf paper.

Minor comments

Line 270 The cited paper from Lennartz et al shows that dblb12 does not bind icam1. DBLβ12 binds the host receptor gC1qR (Magallon Tejada plos pathog 2016),

Line 336 Expression of DC11 was also shown to be upregulated in Papua in severe malaria by RNAseq in 2018 (tonkin-hill et al plos biol 2018)

Fig 5 legend k keo8var -r3 should this read r2?

PLOS authors have the option to publish the peer review history of their article (what does this mean?). If published, this will include your full peer review and any attached files.

Reviewer #1: **Yes: **Reetesh Raj Akhouri

Reviewer #2: **Yes: **Suchi Goel

Reviewer #3: No

Figure Files:

Data Requirements:

Reproducibility:

References:

---

## [Editor Report · Decision Letter 1]

3 Dec 2024

Dear Professor Rowe,

We are pleased to inform you that your manuscript 'Identification of novel PfEMP1 variants containing domain cassettes 11, 15 and 8 that mediate the Plasmodium falciparum virulence-associated rosetting phenotype' has been provisionally accepted for publication in PLOS Pathogens.

Best regards,

Tania F. de Koning-Ward

Academic Editor

PLOS Pathogens

Tracey Lamb

Section Editor

PLOS Pathogens

Sumita Bhaduri-McIntosh

Editor-in-Chief

PLOS Pathogens

orcid.org/0000-0003-2946-9497

Michael Malim

Editor-in-Chief

PLOS Pathogens

orcid.org/0000-0002-7699-2064

The authors have been able to address all but one of the concerns raised by the reviewers.
---

## [Editor Report · Acceptance letter]

3 Jan 2025

Dear Professor Rowe,

We are delighted to inform you that your manuscript, "Identification of novel PfEMP1 variants containing domain cassettes 11, 15 and 8 that mediate the Plasmodium falciparum virulence-associated rosetting phenotype," has been formally accepted for publication in PLOS Pathogens.

Best regards,

Sumita Bhaduri-McIntosh

Editor-in-Chief

PLOS Pathogens

orcid.org/0000-0003-2946-9497

Michael Malim

Editor-in-Chief

PLOS Pathogens

orcid.org/0000-0002-7699-2064